# The free fatty acid receptor GPR164 maintains intestinal homeostasis and barrier function

Takako Ikeda [1,2 ✉], Yuki Masujima[1], Keita Watanabe[1], Akari Nishida[2,6], Mayu Yamano[2], Miki Igarashi[3], Nobuo Sasaki[4], Hironori Katoh [5] & Ikuo Kimura [1,2 ✉]

## Abstract

GPR164 is a free fatty acid receptor, activated by both short-chain fatty acids and medium-chain fatty acids, and expressed throughout the gastrointestinal tract. Although GPR164 is reported to be involved in the release of gut hormones, the physiological functions of this receptor in the maintenance of intestinal homeostasis remain unclear. In this study, we explore the role of GPR164 in regulating intestinal barrier function using mice lacking *Gpr164* gene (*Gpr164*−/−). A loss-of-function mutation in *Gpr164* promotes cell proliferation and disrupts the intestinal barrier function in both Caco-2 cells and mice. Genome-wide RNA-seq analysis reveals that *Gpr164* deletion causes aberrant Wnt/β-catenin signaling, and the intraperitoneal injection of the Wnt/β-catenin inhibitor PNU-74654 ameliorates intestinal hyperproliferation, differentiation and barrier permeability phenotypes of *Gpr164*−/− mice. *Gpr164*−/− mice also exhibit gut microbial dysbiosis and inflammation. Thus, our findings uncover the pivotal role of GPR164 in the maintenance of intestinal homeostasis through regulating barrier function.

**Keywords** Free Fatty Acid Receptor; Short-Chain Fatty Acid; Intestinal Barrier; Microbiota; Wnt Signaling
**Subject Categories** Development; Molecular Biology of Disease; Signal Transduction

## Introduction

GPR164 was initially identified as an olfactory chemosensory receptor. GPR164 is also known as Olfr558 in mouse and OR51E1 in human, and Olfr558 and OR51E1 are encoded by *Or51e1* and *OR51E1* gene, respectively. GPR164 is expressed in several tissues including the olfactory epithelium. In peripheral tissues, GPR164 is activated by free fatty acids (FFAs) such as butyrate and nonanoate, thereby being considered as a free fatty acid receptor (FFAR)

(Bellono et al, 2017; Saito et al, 2009). Although a large number of FFAs such as iso-butyrate, valerate and nonanoate has been identified as ligands for GPR164, physiological abundance of these FFAs is not enough to activate GPR164 in the intestine (Bellono et al, 2017; Halperin Kuhns et al, 2019). Contrary to these FFAs, butyrate is usually present in the colon at millimolar level, and activates GPR164 with an $EC_{50}$ value of 0.51 mM (Halperin Kuhns et al, 2019). Therefore, it is plausible that GPR164 is preferentially activated by butyrate in colon and plays an important role in regulating the intestinal homeostasis. Recently, GPR164 has been reported to have a potential relevance for the enteroendocrine signaling. In pigs, OR51E1 is expressed throughout the gastrointestinal tract, and co-localized with peptide YY (PYY)- and 5-hydroxytryptamine (5-HT)-expressing cells (Priori et al, 2015). Furthermore, in human enteroendocrine L-cell line NCI-H716, OR51E1 is co-localized with glucagon-like peptide-1 (GLP-1), and nonanoate increases the secretion of GLP-1 and PYY through activation of this receptor (Ku et al, 2018). These results suggest an important role for GPR164 in the regulation of gut hormone release. However, the physiological functions of GPR164 in the maintenance of intestinal homeostasis remain largely unclear.

Butyrate is a free fatty acid classified into the short-chain fatty acid (SCFA), and plays an important role in many physiological functions such as energy metabolisms and immune responses (Shimizu et al, 2019; Ikeda et al, 2022). SCFAs are bioactive endproducts produced by the gut microbiota through anaerobic fermentation in the colon (Topping and Clifton 2001). Altered microbial composition (dysbiosis) affects the production of SCFAs, thereby contributing to various diseases such as obesity, type 2 diabetes mellitus (T2DM), and inflammatory bowel disease (IBD). SCFAs are thus crucial mediator linking gut microbiota and optimal host health. In particular, the beneficial role of butyrate in improving the intestinal barrier function is well known. Butyrate enhances the intestinal barrier function by up-regulating the expression of tight junction and mucin proteins, conferring increased protection against enteric toxins and pathogens (Peng et al, 2009; Willemsen et al, 2003). However, the underlying mechanisms of butyrate-mediating maintenance of the intestinal barrier function remain elusive.

[1]Laboratory of Molecular Neurobiology, Graduate School of Biostudies, Kyoto University, Sakyo-ku, Kyoto 606-8501, Japan. [2]Department of Molecular Endocrinology, Graduate School of Pharmaceutical Sciences, Kyoto University, Sakyo-ku, Kyoto 606-8501, Japan. [3]Department of Immunoregulation Institute of Medical Science, Tokyo Medical University, 6-1-1 Shinjuku, Shinjuku-ku, Tokyo 160-8402, Japan. [4]Laboratory of Mucosal Ecosystem Design, Institute for Molecular and Cellular Regulation, Gunma University, Maebashi, Japan. [5]Department of Biological Chemistry, Graduate School of Science, Osaka Metropolitan University, Gakuen-cho, Naka-ku, Sakai, Osaka 599-8531, Japan. [6]Present address: Laboratory of Molecular Neurobiology, Graduate School of Biostudies, Kyoto University, Sakyo-ku, Kyoto 606-8501, Japan. ✉E-mail: ikeda.takako.2r@kyoto-u.ac.jp; kimura.ikuo.7x@kyoto-u.ac.jp

The intestinal epithelial cells self-renew rapidly, which allows for exerting multiple functions involving digestive and absorptive activities (Pint and Clevers 2005). This characteristically rapid turnover relies on the highly proliferative stem cells. Intestinal stem cells in the crypt give rise to transit amplifying cells, which gradually differentiate into other epithelial lineages such as absorptive enterocytes and secretory cells (e.g., enteroendocrine cells, goblet cells). These differentiated cells migrate along the crypt axis, and are removed by apoptosis upon reaching the top of crypt. Wnt/β-catenin signaling pathway (Wnt signaling) regulates cell fate along the crypt axis. Activation of Wnt signaling in the crypt promotes cell proliferation, while attenuation of this signaling in the top of crypt results in cell cycle arrest. Aberrant activation of Wnt signaling initiates and develops colorectal cancers, suggesting an essential role of Wnt signaling in maintaining intestinal homeostasis. Butyrate is also involved in the regulation of intestinal epithelial cell growth through inhibiting histone deacetylase (HDAC) activity (Salvi and Cowles 2021). Although butyrate is rapidly used as fuel in the colon, intake of non-digestive carbohydrates leads to an increased accumulation of butyrate because butyrate is produced by the microbial fermentation of non-digestive carbohydrates as substrates. In addition, the progression of colorectal cancer also increases the level of butyrate (Salvi and Cowles 2021). The colon cancer cells preferentially utilize glucose rather than butyrate as fuel, and the remaining butyrate is accumulated in colon. In both cases, increased butyrate concentration allows butyrate to act as a signaling molecule for FFARs and an HDAC inhibitor. Inhibition of HDAC activity by butyrate resulted in hyperacetylation of histone H3, leading to the transcriptional activation of genes associated with cell cycle inhibition and apoptosis (Salvi and Cowles 2021). Therefore, anti-proliferative property of butyrate through HDAC inhibition has gained attention for the treatment of colorectal cancer.

In this study, we investigated the physiological importance of GPR164 as a key factor for regulating intestinal barrier function using $Gpr164^{-/-}$ mice. The endogenous functions of this receptor remain obscure, and the underlying mechanisms for the maintenance of intestinal homeostasis are not fully understood. Therefore, our findings provide the novel molecular mechanisms for the maintenance of intestinal homeostasis mediated by butyrate/GPR164 axis.

# Results

## Loss of *OR51E1* results in increased cell cycle progression and intestinal barrier dysfunction in Caco-2 cells

In peripheral tissues, GPR164 stimulates adenylyl cyclase in response to FFAs and increases intracellular cAMP levels. First, we performed ligand screening assay using Or51e1-overexpressing HEK293 cells to identify the endogenous ligands. To functionally express transfected Or51e1 on the cell surface, a receptor-transporting protein (RTP) was co-transfected with Or51e1 because GPR164 is an olfactory receptor that requires the chaperones such as RTPs for the ectopic expression in other non-nasal tissues (Zhuang and Matsunami 2007). In mammals, RTP family consists of 4 members (RTP1-4), and a shorter form of RTP1 (RTP1S) was reported to promote cell-surface expression of GPR164 (Halperin

Kuhns et al, 2019). However, mRNA expression levels of *Rtp1S*, *Rtp2* and *Rtp3* in mouse colon were lower than that of *Rtp4* (Fig. 1A). We therefore performed ligand screening by using cells expressing either RTP1S or RTP4. The Or51e1 expression was confirmed by transient co-transfection with either RTP1S or RTP4 in HEL293 cells (Fig. EV1A), and intracellular cAMP levels were upregulated by the treatment with butyrate or valerate (Fig. 1B). It is interesting to note that decanoate increased cAMP level only when cells were co-expressed with RTP1S (Fig. 1B). We next investigated the functions of GPR164 using *OR51E1* knockout Caco-2 (*OR51E1* KO) cells generated by the CRISPR/Cas9 system (Fig. EV1B). GPR164 was previously reported to induce cell cycle arrest and cell death (Pronin and Slepak 2021). Therefore, we examined the mRNA expression levels of genes associated with cell cycle regulation, and found the decreased expression of *p21* and *p27* in *OR51E1* KO cells (Fig. 1C). Additionally, the mRNA expressions of cyclin genes in *OR51E1* KO cells were higher than that in control cells at day 18 (Fig. 1C). Consistent with these results, *OR51E1* KO cells exhibited increased cell growth after day 15, but not until day 12 when compared to control cells (Fig. 1D), suggesting that GPR164 is a key regulator of cell proliferation. Caco-2 cells, which are derived from a colon carcinoma, form a polarized epithelial cell monolayer after reaching confluence, and are used as a model of the intestinal epithelial barrier. In *OR51E1* KO cells, mRNA expression of *Occludin* and *Claudin-3* was decreased, whereas *Zo-1* mRNA expression was increased after confluence compared with control cells (Fig. 1E). In addition, reduction in transepithelial electrical resistance (TER), a reliable in vitro measurement of barrier function, was observed in *OR51E1* KO cells after day 15 (Fig. 1F). Although GPR164 was reported to induce p53-dependent cell cycle arrest and apoptosis (Pronin and Slepak 2021), we could not detect a change in p53 protein expression between control and *OR51E1* KO cells (Figs. 1G and EV1C).

## *Gpr164*⁻/⁻ mice exhibit colonic hyperplasia and impaired barrier function

Next, we investigated the physiological expression levels of *Or51e1* in the mouse intestine and olfactory epithelium. Consistent with previous reports (Bellono et al, 2017), *Or51e1* was expressed not only in the olfactory epithelium but also in the intestine (Fig. 2A). Although *Or51e1* expresses throughout the intestine, higher expression was observed in the colon (Fig. 2A). We therefore generated *Or51e1* gene knockout (*Gpr164*⁻/⁻) mice by using the CRISPR/Cas9 system in wild-type C57BL/6 J zygotes (Fig. EV1D–F). To investigate the morphological changes in the colon of male *Gpr164*⁻/⁻ mice, hematoxylin and eosin (HE) staining was performed by using paraffin-embedded cross-section of colon. HE staining revealed increased area of the colon in *Gpr164*⁻/⁻ mice (Fig. 2B), and whole length of the colon and the depth of colonic crypt in *Gpr164*⁻/⁻ mice were longer than those in WT mice (Fig. 2C,D). To examine whether the colonic hyperplasia observed in *Gpr164*⁻/⁻ mice was caused by aberrant cell cycle progression, we examined the expression of cell cycle genes and Ki67 cell proliferation marker protein. In *Gpr164*⁻/⁻ mice, the mRNA expression of *p21* and *p27* was decreased, but that of cyclin genes was increased (Fig. 2E). Immunohistochemical analysis of Ki67 reveled enhanced cell proliferation in *Gpr164*⁻/⁻ mice (Fig. 2F), suggesting that accelerated cell proliferation caused by *Gpr164*

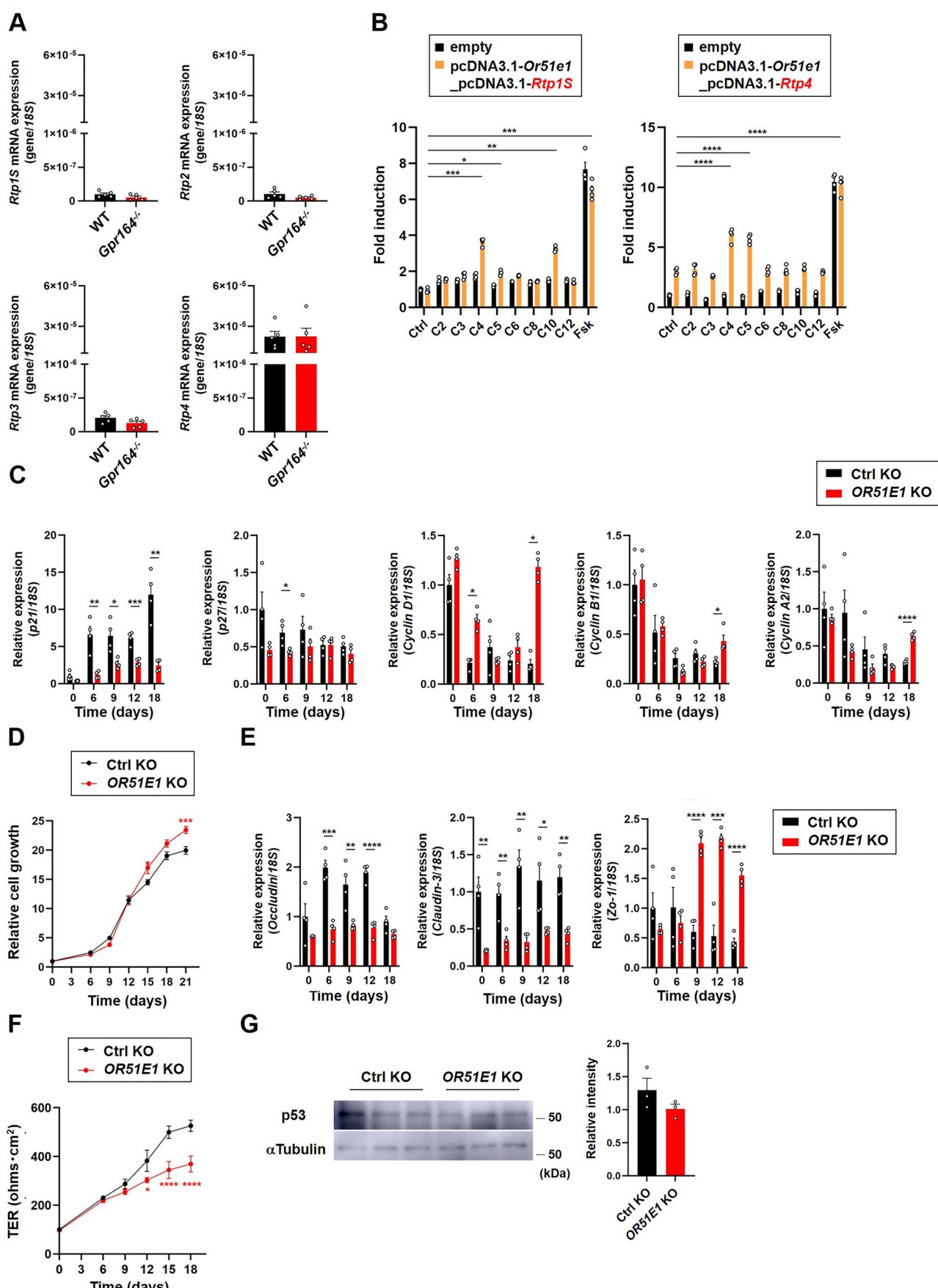

**Figure 1. OR51E1-knockout Caco-2 cells exhibit intestinal barrier dysfunction.**

(A) The mRNA expression levels of RTP family members determined by qRT-PCR. Total RNA was extracted from colon tissue of WT and $Gpr164^{-/-}$ mice ($n = 5$). Error bars represent the mean ± SEM. (B) Measurement of intracellular cAMP in Or51e1-overexpressing HEK293 cells. Cells were co-transfected with HA-tagged Or51e1 and receptor-transporting protein (left; RTP1S, right; RTP4), and treated with the indicated FFA at 1 mM for 10 min. Folskolin (FSK) was used as a positive control. The values represent the mean ± SEM from two independent experiments ($n = 4$). ***$P = 0.0002$; Ctrl vs C4 (left), *$P = 0.0181$; Ctrl vs C5 (left), **$P = 0.0012$; Ctrl vs C10 (left), ****$P < 0.0001$; Ctrl vs FSK (left), ****$P < 0.0001$; Ctrl vs C4 (right), ****$P < 0.0001$; Ctrl vs C5 (right), ****$P < 0.0001$; Ctrl vs FSK (right) (Dunn's test; left, Dunnett's test; right). (C) The mRNA expression levels of cell cycle genes determined by qRT-PCR. The bars represent the relative mRNA levels from at least three independent experiments ($n = 4$). Error bars show the mean ± SEM. **$P = 0.0028$; $p21$ gene at day 6, *$P = 0.0152$; $p21$ gene at day 9, ***$P = 0.0006$; $p21$ gene at day 12, **$P = 0.0013$; $p21$ gene at day 18, *$P = 0.0161$; $p27$ gene at day 6, *$P = 0.0286$; Cyclin D1 gene at day 6, *$P = 0.0286$; Cyclin D1 gene at day 18, *$P = 0.0146$; Cyclin B1 gene at day 18, ****$P < 0.0001$; Cyclin A2 gene at day 18 (Cyclin D1 gene; Mann–Whitney $U$-test, other genes; Student's $t$-test). (D) Measurement of cell proliferation by crystal violet staining. Cultured cells were stained with crystal violet at the indicated times, and absorbance was measured at OD$_{590}$ nm. The values represent the mean ± SEM from three independent experiments ($n = 6$). ***$P = 0.0008$; Ctrl KO vs OR51E1 KO at day 21 (Sidak's test). (E) The mRNA expression levels of tight junction markers determined by qRT-PCR. The bars represent the relative mRNA levels from at least three independent experiments ($n = 4$). Error bars show the mean ± SEM. ***$P = 0.0003$; Occludin gene at day 6, **$P = 0.0097$; Occludin gene at day 9, ****$P < 0.0001$; Occludin gene at day 12, **$P = 0.0073$; Claudin-3 gene at day 0, **$P = 0.0052$; Claudin-3 gene at day 6, **$P = 0.004$; Claudin-3 gene at day 9, *$P = 0.0308$; Claudin-3 gene at day 12, **$P = 0.0017$; Claudin-3 gene at day 18, ****$P < 0.0001$; Zo-1 gene at day 9, ***$P = 0.0002$; Zo-1 gene at day 12, ****$P < 0.0001$; Zo-1 gene at day 18 (Student's $t$-test). (F) Monitoring of TER at different times. The values represent the mean ± SEM from at least three independent experiments ($n = 3$). *$P = 0.0175$; Ctrl KO vs OR51E1 KO at day 12, ****$P < 0.0001$; Ctrl KO vs OR51E1 KO at day 15 and 18 (Sidak's test). (G) Representative image of p53 protein expression. Total cell lysates extracted from Ctrl KO or OR51E1 KO cells were subjected to immunoblot analysis using an anti-p53 or anti-αTubulin antibody. The bars represent the relative intensity of p53 bands ($n = 3$). Error bars show the mean ± SEM. Source data are available online for this figure.

deficiency leads to the colonic hyperplasia. $Gpr164^{-/-}$ female mice also exhibited the hyperplasia phenotype in the colon such as an increased volume and cell cycle progression (Fig. EV2A–C). Furthermore, decreased expression of tight junction marker genes was observed in both male and female $Gpr164^{-/-}$ mice (Figs. 2G and EV2C). We thus assessed the role of GPR164 in the intestinal barrier function in vivo by using fluorescein isothiocyanate (FITC) labeled-dextran. After oral administration of FITC-dextran, plasma level of FITC-dextran in $Gpr164^{-/-}$ mice was elevated (Fig. 2H). To confirm the effects of GPR164 on cell proliferation and the intestinal barrier, we also demonstrated 5-bromo-2'-deoxyuridine (BrdU) incorporation assay and lactulose/mannitol test. BrdU immunohistochemistry demonstrated the increased BrdU incorporation in $Gpr164^{-/-}$ mice, suggesting an enhanced progression of DNA synthesis during mitosis (Fig. EV3A). In addition to accelerated proliferation of the intestinal epithelial cells, reduced cell death such as apoptosis may be a cause of colonic hyperplasia in $Gpr164^{-/-}$ mice. We thus performed immunofluorescence staining for cleaved caspase-3, but found no apparent differences between genotypes in apoptosis (Fig. EV3B). For the assessment of intestinal permeability, we measured urinary lactulose/mannitol (Lac/Man) ratio. The Lac/Man ratio was increased in $Gpr164^{-/-}$ mice when compared to WT mice (Fig. EV3C). Hence, these observations indicate that GPR164 plays an essential role in the regulation of cell growth and the maintenance of intestinal barrier function.

## Loss of Gpr164 induces gut microbial dysbiosis and intestinal inflammation

The mucus layer, which is composed of complex glycans including Muc2 mucin $O$-glycan, serves as a physical barrier against luminal pathogens in the colon (Johansson et al, 2010). To investigate whether the impairment of the intestinal barrier function observed in $Gpr164^{-/-}$ mice is caused by the defects in the colon mucus, alcian blue-periodic acid Schiff (PAS) staining was performed. The colon section from $Gpr164^{-/-}$ mice was strongly stained with alcian blue-PAS, but the mucus layer of $Gpr164^{-/-}$ mice was thinner than

that of WT mice (Fig. 3A,B). The gut microbiota inhabits the colon mucus which separates bacteria and host (physical barrier) and contains anti-bacterial proteins responsible for blocking the bacteria from penetrating the epithelial cell barrier (immune barrier). Therefore, impaired mucus barrier function alters the microbial community and induces colonic inflammation (Swidsinski et al, 2007). Given that GPR164 plays an important role in the mucus barrier function, we examined the alterations in gut microbial composition by using fecal samples of WT or $Gpr164^{-/-}$ mice. 16S rRNA sequencing showed that the microbial composition of $Gpr164^{-/-}$ mice shifted toward that of WT mice fed high-fat diet (HFD), a well-known risk factor for metabolic and inflammatory diseases induced by the gut microbial dysbiosis (Fig. 3C). The composition of microbiota in $Gpr164^{-/-}$ mice fed HFD was also different from that in either WT mice fed HFD or $Gpr164^{-/-}$ mice fed ND (Fig. 3C). The hierarchical clustering analysis confirmed marked alterations in the microbial composition, and qRT-PCR analysis revealed the increased abundance of Firmicutes in $Gpr164^{-/-}$ mice (Fig. 3D,E). The Firmicutes/Bacteroides (F/B) ratio, a biomarker of gut microbial dysbiosis, was higher in $Gpr164^{-/-}$ mice than in WT mice under HFD feeding conditions (Fig. 3E). The gut microbial dysbiosis is strongly associated with inflammation, and we then examined the involvement of GPR164 in inflammation. Notably, plasma level of TNFα in $Gpr164^{-/-}$ mice was comparable to that observed in WT mice fed HFD, and $Gpr164^{-/-}$ mice exhibited inflammation even under normal diet conditions (Fig. 3F). We further examined the impact of intestinal dysbiosis on the production of microbial metabolites, and found the decreased levels of butyrate in the colon of $Gpr164^{-/-}$ mice (Fig. 3G). Taken together, these results indicate that GPR164 is involved in the regulation of the mucus barrier, microbial composition, SCFA production and inflammation.

## Gpr164$^{-/-}$ mice have defects in the intestinal epithelial lineage caused by aberrant Wnt signaling

Mucins are condensed in the secretory granules of goblet cells. Considering that the colon sections from $Gpr164^{-/-}$ mice were

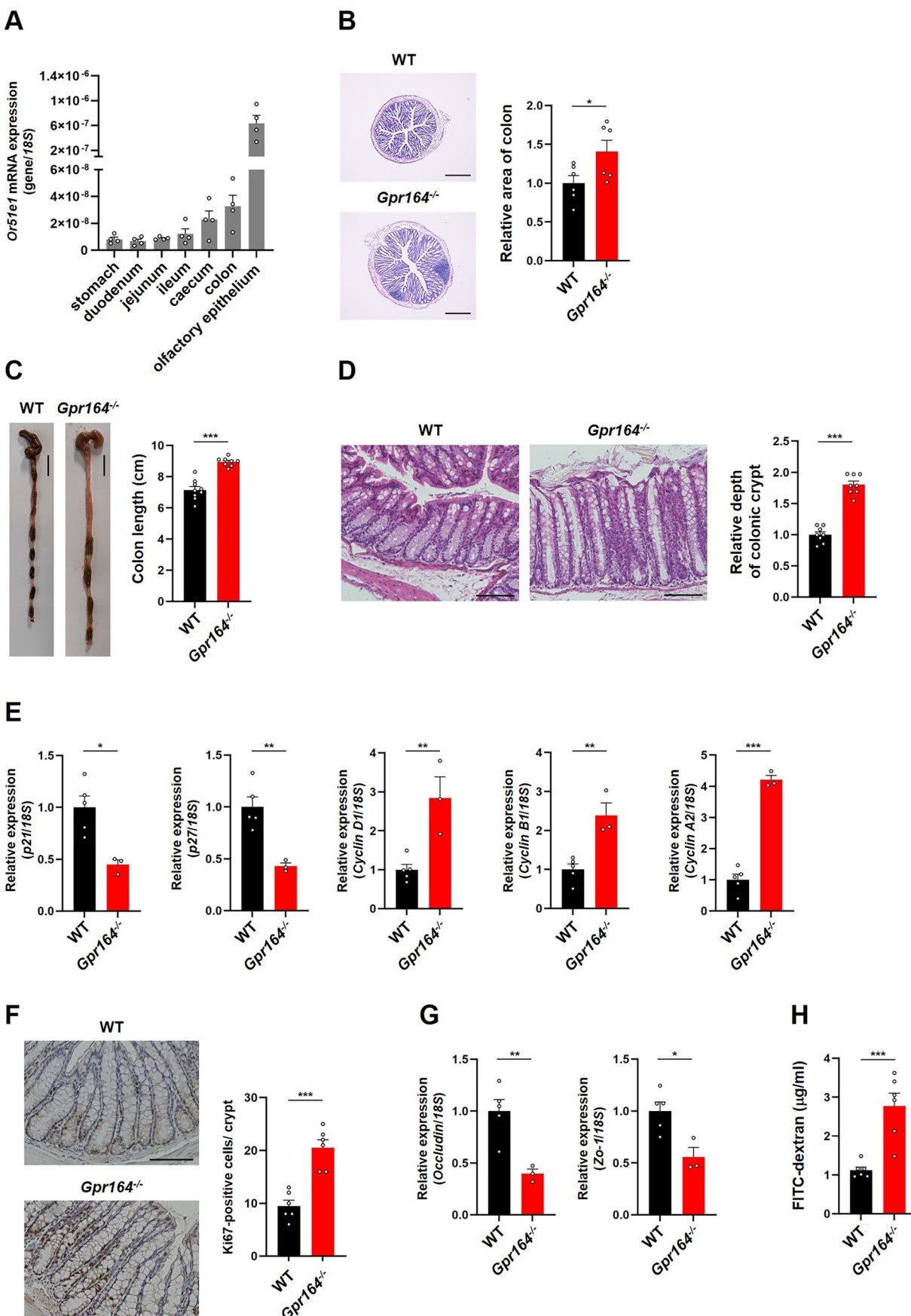

**Figure 2. *Gpr164*<sup>−/−</sup> mice exhibit colonic hyperplasia and intestinal barrier dysfunction.**

(A) *Or51e1* mRNA expression determined by qRT-PCR. Total RNA was extracted from gastrointestinal tract of WT mice ($n = 4$). Error bars represent the mean ± SEM.
(B) Representative images of colon sections stained with hematoxylin-eosin (HE). The cross-sections of colon obtained from WT and *Gpr164*<sup>−/−</sup> mice were stained with HE, and the area of colon was measured using ImageJ software ($n = 6$). Scale bar, 500 µm. Error bars represent the mean ± SEM. *$P = 0.0434$ (Student's *t*-test).
(C) Representative images of colon ($n = 8$–9). Scale bar, 1 cm. Error bars represent the mean ± SEM. ****$P < 0.0001$ (Student's *t*-test). (D) The colon sections stained with HE. The depth of the colonic crypt was measured using ImageJ software ($n = 8$). Scale bar, 100 µm. Error bars represent the mean ± SEM. ****$P < 0.0001$ (Student's *t*-test).
(E) The mRNA expression levels of cell cycle genes determined by qRT-PCR. Total RNA was extracted from colon tissue of WT and *Gpr164*<sup>−/−</sup> mice ($n = 3$–5). Error bars represent the mean ± SEM. *$P = 0.0102$; *p21* gene, **$P = 0.0048$; *p27* gene, **$P = 0.0055$; *Cyclin D1* gene, **$P = 0.0037$; *Cyclin B1* gene, ****$P < 0.0001$; *Cyclin A2* gene (Student's *t*-test). (F) Representative images of immunohistochemical staining for Ki67 in colon. The cross-sections of colon obtained from WT and *Gpr164*<sup>−/−</sup> mice were stained with anti-Ki67 antibody. DAB was used for the detection of Ki67 protein, and hematoxylin was used for nuclei staining ($n = 6$). Scale bar, 100 µm. Error bars represent the mean ± SEM. ***$P = 0.0002$ (Student's *t*-test). (G) The mRNA expression levels of tight junction markers determined by qRT-PCR. Total RNA was extracted from colon tissue of WT and *Gpr164*<sup>−/−</sup> mice ($n = 3$–5). Error bars represent the mean ± SEM. **$P = 0.0076$; *Occludin* gene, *$P = 0.0163$; *Zo-1* gene (Student's *t*-test).
(H) Assessment of intestinal permeability by measuring plasma levels of FITC-dextran. The fluorescence intensity of FITC-dextran in plasma was measured ($n = 6$). Error bars represent the mean ± SEM. ***$P = 0.0007$ (Student's *t*-test). Source data are available online for this figure.

strongly stained with alcian blue-PAS (Fig. 3A), it may be caused by an increased number of goblet cells. The colonic progenitor cells generated from Lgr5-expressing stem cells gradually differentiate into the two main epithelial lineages: absorptive and secretory lineage. The Hes1-expressing absorptive progenitors differentiate into all enterocytes, while Atoh1-expressing secretory progenitors differentiate into either enteroendocrine cells or goblet cells. We thus examined the mRNA expression of epithelial lineage markers, and found that the expression of *Lgr5* was decreased but that of *Atoh1* was increased in colon of *Gpr164*<sup>−/−</sup> mice (Fig. 4A). To further elucidate the role of GPR164 in intestinal epithelial differentiation, we performed genome-wide RNA sequencing by using colon samples from WT and *Gpr164*<sup>−/−</sup> mice. KEGG enrichment analysis revealed that the expression of genes related to the regulation of receptor activities were significantly changed in *Gpr164*<sup>−/−</sup> mice (Fig. EV4A). Notably, the expression of marker genes for enteroendocrine cells (*Chga*, *Gcg*, and *Pyy*) and absorptive cells (*Slc26a2* and *Slc4a2*) were reduced, whereas these for goblet cells (*Reg4*, *Spink4*, and *Fcgbp*) were elevated in *Gpr164*<sup>−/−</sup> mice (Fig. 4B). In addition, quantitative PCR analysis showed that the expression of *Tph-1* and *Vil1* was decreased but that of *Muc2* was increased in both male and female *Gpr164*<sup>−/−</sup> mice, confirming that a loss of *Gpr164* resulted in the enhanced differentiation into goblet cells (Figs. 4B and EV2C). The proliferation and differentiation of colonic epithelial cells are regulated by Wnt/β-catenin signaling pathway. Upon activation of Frizzled and LRP5/6 receptors by binding Wnt ligands, β-catenin escapes from proteasomal degradation and accumulates in the nucleus, where it promotes transcription of genes involved in cell cycle progression by interacting with transcription factors of the T-cell factor/lymphoid enhancing factor (TCF/LEF) family (Fig. 4C). Therefore, Wnt signaling is essential for maintaining a proliferative phenotype of epithelial stem/progenitor cells in crypt, whereas the absence of this signal in the top of crypt is important to allow epithelial cell differentiation. A heatmap of the gene expression profile revealed that the expression levels of many genes associated with Wnt signaling pathway were changed markedly, and aberrant expression of β-catenin in the top of crypt was observed in *Gpr164*<sup>−/−</sup> mice (Fig. 4C,D). Besides, nuclear localization of β-catenin in the epithelial cells of *Gpr164*<sup>−/−</sup> mice was also observed (Fig. 4D, right). Thus, these findings suggest that *Gpr164* deficiency results in both aberrant expression and ectopic localization of β-catenin in the epithelial cells, which causes abnormal proliferation (colonic hyperplasia) and differentiation

(increased goblet cells and reduced enteroendocrine/absorptive cells) of colonic epithelial cells. On the contrary, mRNA expression of *Lgr5* was reduced in *Gpr164*<sup>−/−</sup> mice, implying the possibility that colonic epithelial stem cells in the crypt cannot be received an appropriate Wnt signaling or the proportion of Lgr5 positive stem cells was reduced because of the increased number of total epithelial cells. It is well-known that dysregulation of Wnt signaling results in several diseases such as IBD and colorectal cancers (Aust et al, 2002; Sparks et al, 1998). Consistent with the data obtained in Figs. 2H and 3F, Kyoto Encyclopedia of Genes and Genome (KEGG) analysis showed that *Gpr164* deletion leads to disruption of epithelial barrier integrity and promotion of inflammation in the colon (Fig. EV4B,C). In *Gpr164*<sup>−/−</sup> mice, many genes involved in the tight junction assembly and cell polarity were reduced, whereas several marker genes for effector T cells were increased (Fig. EV4B,C). To confirm findings of the data shown in Fig. EV4C, we performed immunofluorescent staining for CD4 and found an increase in CD4 positive cells in the colon lamina propria of *Gpr164*<sup>−/−</sup> mice (Fig. EV4D).

## Inhibition of Wnt signaling ameliorates abnormalities in the colon of *Gpr164*<sup>−/−</sup> mice

To further elucidate the functional role of GPR164 in maintaining intestinal homeostasis, we examined the effect of Wnt inhibitor, PNU-74654, on the development of colonic epithelial abnormalities caused by *Gpr164* deletion. PNU-74654 acts as a Wnt/β-catenin antagonist by preventing TCF from binding to β-catenin, and inhibits tumor growth in a mouse model of colorectal cancer (Amerizadeh et al, 2022). Intraperitoneal administration of PNU-74654 suppressed the hypertrophic phenotypes in the colon of *Gpr164*<sup>−/−</sup> mice (Fig. 5A,B). The anti-proliferative effects of PNU-74654 in *Gpr164*<sup>-/-</sup> mice were mediated through repressing increased expressions of *cyclin D1* and *c-Myc*, although decreased expression of *p21* and *p27* were partially restored (Figs. 5C and EV5). Also, a marked reduction in Ki67 protein expression of *Gpr164*<sup>−/−</sup> mice confirmed anti-proliferative effect of PNU-74654 (Fig. 5D), suggesting that accelerated proliferation in *Gpr164*<sup>−/−</sup> mice is caused by Wnt signaling overactivation. Next, we investigated the involvement of Wnt signaling in abnormal differentiation of colonic epithelial cells in *Gpr164*<sup>−/−</sup> mice. Upon treatment with PNU-74654, a lower level of *Vil1* and a higher level of *Muc2* expression in *Gpr164*<sup>−/−</sup> mice were restored to similar level

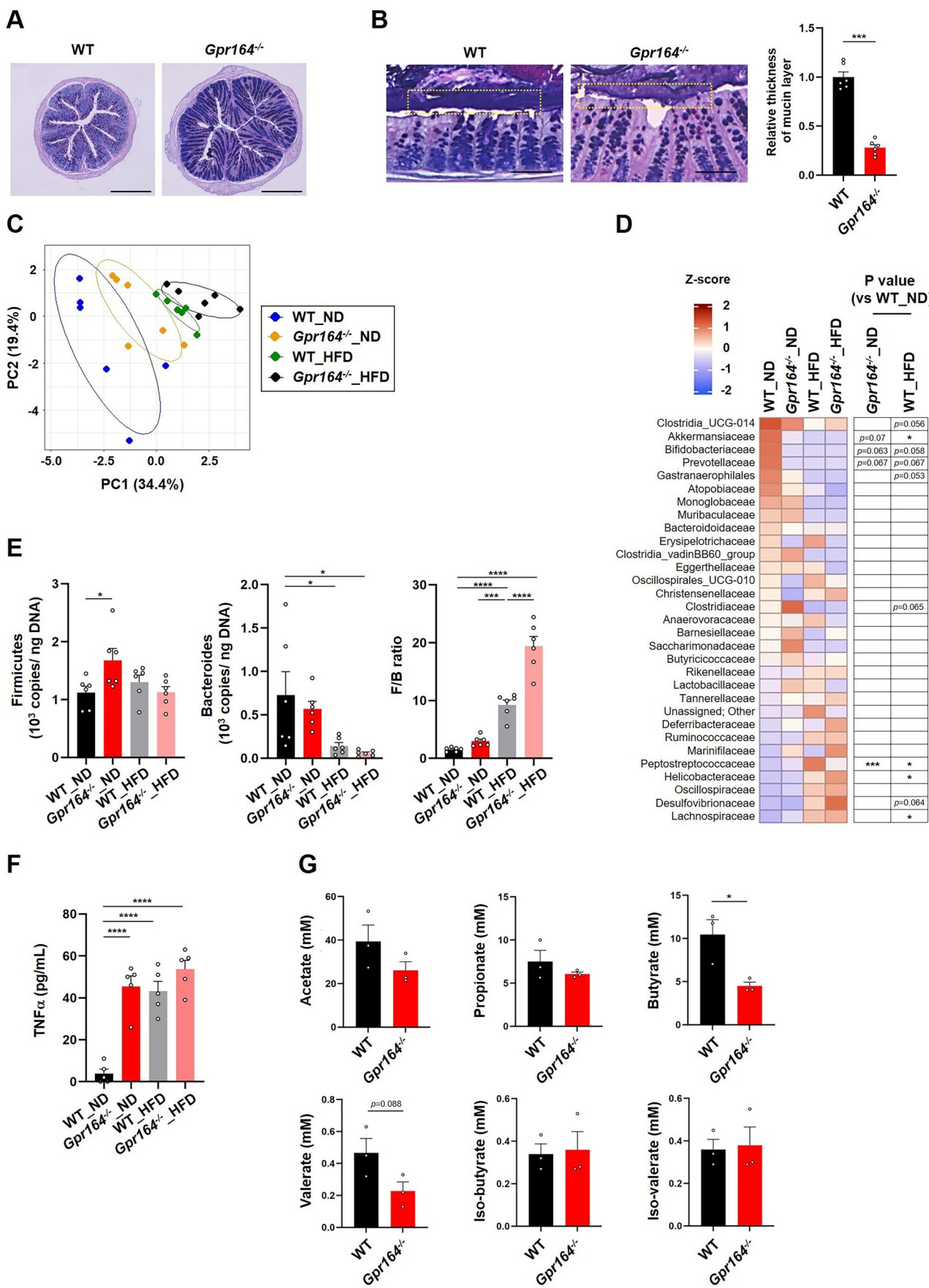

Figure 3.  Loss of *Or51e1* results in gut microbial dysbiosis and inflammation.

(A, B) Representative images of colon sections stained with alcian blue-periodic acid Schiff (PAS). Low- (A) or high- (B) magnification cross-sectional images of colon obtained from WT and Gpr164$^{-/-}$ mice are shown ($n = 6$). A part of mucus layer was bounded by yellow dotted lines (B). Relative thickness of mucin layer was measured using ImageJ software ($n = 6$). Error bars represent the mean ± SEM. ****$P < 0.0001$ (Student's *t*-test). Scale bar, 500 µm (A) and 100 µm (B). (C) Principal coordinate analysis of fecal microbiota from WT and Gpr164$^{-/-}$ mice fed either a normal diet (ND) or a high-fat diet (HFD) for 10 weeks ($n = 6$). (D) Heatmap of relative abundance of taxonomic units in the gut microbiota at the family level ($n = 6$) (Student's *t*-test). (E) The abundance of Firmicutes and Bacteroides in the gut microbiota (left), and Firmicutes/ Bacteroidota (F/B) ratio (right) ($n = 6$). Error bars represent the mean ± SEM. *$P = 0.046$; WT_ND vs Gpr164$^{-/-}$_ND of Firmicutes, *$P = 0.0439$; WT_ND vs WT_HFD of Bacteroides, *$P = 0.0185$; WT_ND vs Gpr164$^{-/-}$_HFD of Bacteroides, ***$P = 0.001$; WT_HFD vs Gpr164$^{-/-}$_ND of F/B ratio, ****$P < 0.001$ (Tukey–Kramer test). (F) Plasma levels of TNFα in WT and Gpr164$^{-/-}$ mice fed either a ND or a HFD for 10 weeks ($n = 5$). Error bars represent the mean ± SEM. ****$P < 0.0001$ (Tukey–Kramer test). (G) SCFA levels in colonic contents ($n = 3$). Error bars represent the mean ± SEM. *$P = 0.0287$; Butyrate, $P = 0.088$; Valerate (Student's *t*-test). Source data are available online for this figure.

in WT mice (Fig. 5E), and strong staining for mucin in the colon section of *Gpr164*$^{-/-}$ mice was attenuated (Fig. 5F). Therefore, these results suggest that aberrant Wnt signaling caused by *Gpr164* deletion leads to the pathophysiologic consequences such as abnormal differentiation in colonic epithelial cells. To examine whether dysregulation of Wnt signaling in *Gpr164*$^{-/-}$ mice affects the intestinal barrier function, mRNA expression of tight junction markers, and plasma TNFα levels were determined. The reduced mRNA expression of *Occludin* and *Zo-1* and high levels of plasma TNFα were rescued to WT levels by administering PNU-74654 to *Gpr164*$^{-/-}$ mice (Fig. 5G,H). Together, these results suggest that GPR164 plays an important role in Wnt signaling-mediated colonic epithelial turnover.

## Butyrate enhances epithelial barrier function through GPR164 activation

Butyrate is a bioactive product generated through microbial fermentation, and strengthens both tight junction and mucus barrier functions. Given that butyrate is a dominant ligand for GPR164 in colon, butyrate-mediated enhancement of the intestinal barrier function may be exerted through GPR164 activation. We assessed this possibility by using colonic organoid isolated from WT and *Gpr164*$^{-/-}$ mice. The intestinal organoid is a 3D in vitro model with self-renewal capacity, and shows similar functionality to the tissue origin (Sasaki and Clevers 2018). The colonic epithelial organoids isolated from *Gpr164*$^{-/-}$ mice showed no obvious differences compared to those from WT mice, when maintained in culture without passage (Fig. 6A). Interestingly, organoids from *Gpr164*$^{-/-}$ mice exhibited crypt-like budding structures after a single passage (Fig. 6A). The budding organoids possess similar features to the mature intestine (Onozato et al, 2021), thereby indicating a possibility that loss-of-function mutation in *Gpr164* is associated with accelerated differentiation and maturation of colonic epithelial cells. Consistent with in vivo findings, mRNA expression level of *Vil1* and *Occludin* was reduced but that of *Muc2* was increased in *Gpr164*$^{-/-}$ organoids (Fig. 6B). In addition, expressions of cytostatic genes, *p21* and *p27*, were decreased although no significant changes in *Cyclin D1* and *c-Myc* mRNA expression were observed (Fig. 6B). Finally, we investigated the effect of butyrate on the intestinal barrier function. For the measurement of the organoid permeability, the luminal FITC-dextran which moves from the basolateral surface was examined (Hu et al, 2023). FITC fluorescence in the lumen was detected in *Gpr164*$^{-/-}$ organoids treated with or without butyrate (Fig. 6C). Upon stimulation with palmitate, well-known as lipotoxic FFA (Ohue-Kitano et al, 2023), increasing intensity of luminal FITC fluorescence

was observed in both WT and *Gpr164*$^{-/-}$ organoid, and pretreatment with butyrate suppressed palmitate-induced increase in luminal FITC fluorescence in WT organoid but not in *Gpr164*$^{-/-}$ organoid (Fig. 6C). Palmitate has been reported to alter the expression of Hes1 and Muc2, thereby influencing the differentiation of certain types of cells such as goblet cells (Filippello et al, 2022). Therefore, besides pro-inflammatory property (Ohue-Kitano et al, 2023), palmitate induces lipotoxicity through impairment of epithelial differentiation. We also examined the beneficial effect of butyrate on epithelial barrier function in Caco-2 cells. A decreased expression level of *Claudin-3* mRNA in palmitate-treated control KO cells tended to be restored by pretreatment of butyrate (Fig. 6D). However, butyrate failed to restore *Claudin-3* expression in *OR51E1* KO cells treated with palmitate (Fig. 6D). Furthermore, palmitate decreased TER in both control and *OR51E1* KO cells, and butyrate suppressed palmitate-induced increased permeability only in control KO cells (Fig. 6E). Taken together, our results indicate that butyrate-mediated activation of GPR164 plays a crucial role in protecting the barrier function.

## Discussion

In the present study, we revealed the pivotal role of a novel FFAR, GPR164, on the intestinal barrier function using *Gpr164*$^{-/-}$ mice. To date, there have been no reports describing in vivo functions of GPR164 by using *Gpr164*$^{-/-}$ mice. Thus, our study provides the first line of evidence how GPR164 maintains the physiological home-ostasis. In this study, we performed ligand screening assay by using HEK293 cells which were co-transfected with both Or51e1 and RTPs (Fig. 1B). Consistent with a previous report (Halperin Kuhns et al, 2019), GPR164 was activated by the stimulation with butyrate or decanoate when cells were co-transfected with both Or51e1 and RTP1S (Fig. 1B). This result indicates that GPR164 is a receptor for both short-chain and medium-chain fatty acid. However, we found much lower level of *Rtp1S* mRNA than that of *Rtp4* in the colon (Fig. 1A), suggesting that RTP4 is a dominant chaperone required for GPR164 trafficking to the plasma membrane in the colon. Interestingly, only SCFAs (butyrate and valerate) activated GPR164 in cells expressing RTP4 (Fig. 1B). Therefore, these results suggest that SCFAs, especially butyrate, play central roles in regulating biological processes through GPR164 activation in the colon.

We here showed that loss of *Gpr164* resulted in the intestinal barrier dysfunction in both in vitro and in vivo models (Figs. 1F, 2H and EV3C). In addition to this, the enhanced proliferation of the intestinal epithelial cells was observed in both models (Figs. 1D, 2F and EV3A), and *Gpr164*$^{-/-}$ mice exhibited the colonic

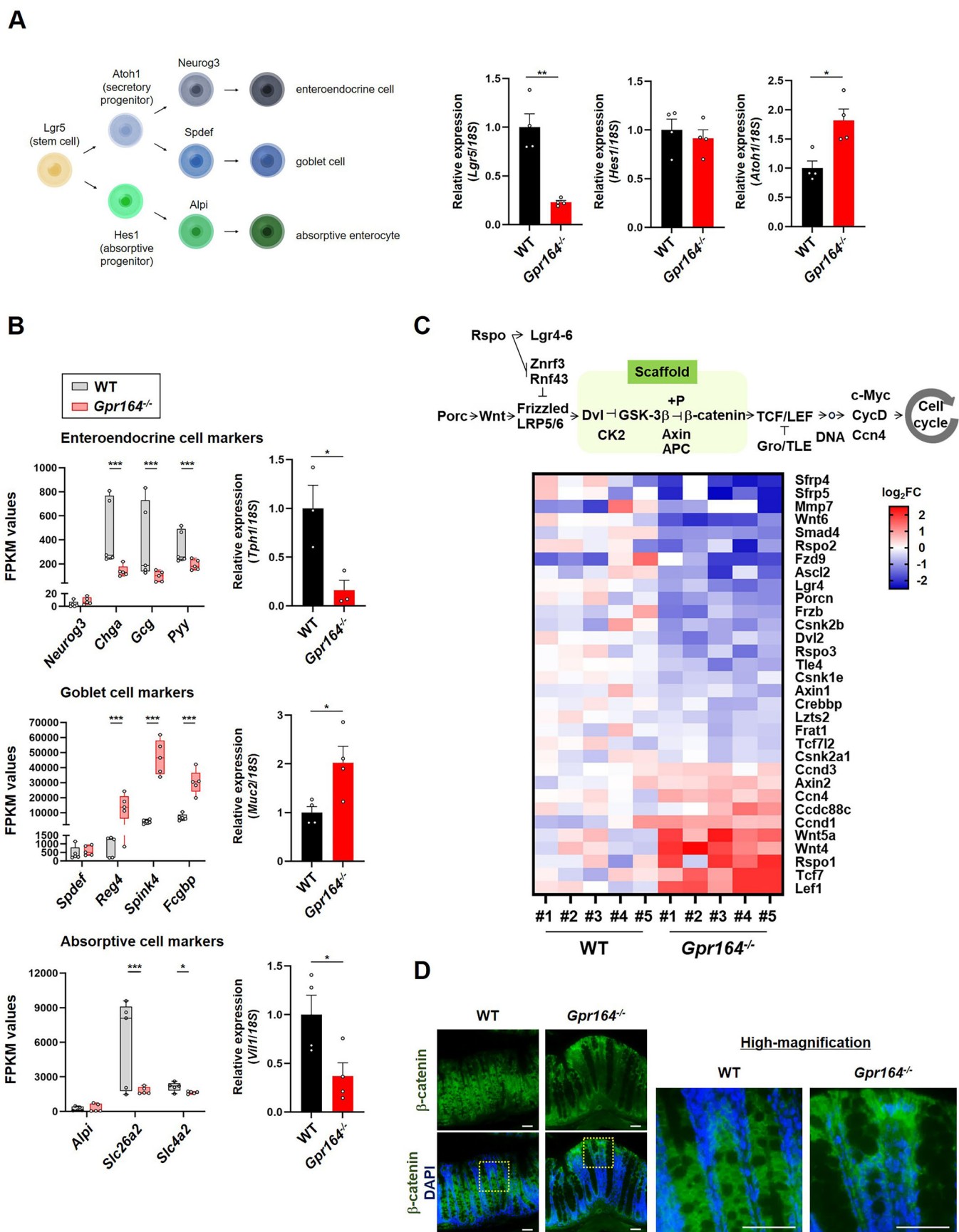

**Figure 4.  Aberrant Wnt signaling causes dysregulation of colonic epithelial differentiation in *Gpr164⁻/⁻* mice.**

(A) Schematic illustration of the intestinal epithelial lineage (left), and the mRNA expression levels of *Lgr5*, *Hes1* and *Atoh1* determined by qRT-PCR. Total RNA was extracted from colon tissue of WT and *Gpr164⁻/⁻* mice (n = 4). Error bars represent the mean ± SEM. **P = 0.0014; *Lgr5* gene, *P = 0.0125; *Atoh1* gene (Student's *t*-test).
(B) FPKM values of genes expressed in enteroendocrine cells (left, upper), goblet cells (left, middle) and absorptive cells (left, bottom). The mRNA expression levels of *Tph1* (right, upper), *Muc2* (right, middle) and *Vil1* (right, bottom) determined by qRT-PCR. Total RNA was extracted from colon tissue of WT and *Gpr164⁻/⁻* mice (n = 3–4). Error bars represent the mean ± SEM. *P = 0.0368; *Chga* gene, *P = 0.0487; *Pyy* gene, *P = 0.031; *Tph1* gene, *P = 0.0113; *Reg4* gene, ****P < 0.0001; *Spink4* gene, ***P = 0.0002; *Fcgbp* gene, *P = 0.0289; *Muc2* gene, *P = 0.0435; *Slc26a2* gene, *P = 0.0251; *Slc4a2* gene, *P = 0.04; *Vil1* gene (Student's *t*-test). *Neurog3*; WT (Max: 11.65, Min: 0, Median: 1.7, 75th percentile: 7.24, 25th percentile: 0), *Gpr164⁻/⁻* mice (Max: 16.48, Min: 2.84, Median: 6.49, 75th percentile: 14.11, 25th percentile: 3.98). *Chga*; WT (Max: 807.74, Min: 239.43, Median: 272.54, 75th percentile: 766.67, 25th percentile: 245.85), *Gpr164⁻/⁻* mice (Max: 218.53, Min: 99.15, Median: 129.23, 75th percentile: 177.65, 25th percentile: 106.24). *Gcg*; WT (Max: 832.17, Min: 125.64, Median: 187.49, 75th percentile: 730.39, 25th percentile: 137.75), *Gpr164⁻/⁻* mice (Max: 148.72, Min: 50.26, Median: 101.3, 75th percentile: 138.36, 25th percentile: 51.01). *Pyy*; WT (Max: 517.44, Min: 226.48, Median: 257.91, 75th percentile: 490.69, 25th percentile: 234.16), *Gpr164⁻/⁻* mice (Max: 247.66, Min: 147.02, Median: 178.97, 75th percentile: 236.83, 25th percentile: 151.63). *Spdef*; WT (Max: 1135.64, Min: 193.1, Median: 262.24, 75th percentile: 785.66, 25th percentile: 211.77), *Gpr164⁻/⁻* mice (Max: 947.19, Min: 324.97, Median: 506.16, 75th percentile: 907.37, 25th percentile: 339.24). *Reg4*; WT (Max: 1363.43, Min: 184.47, Median: 1256.04, 75th percentile: 1344.47, 25th percentile: 189.67), *Gpr164⁻/⁻* mice (Max: 24238.55, Min: 834.16, Median: 13693.45, 75th percentile: 20854.74, 25th percentile: 6039.09). *Spink4*; WT (Max: 5712.5, Min: 2295.58, Median: 4052.5, 75th percentile: 5336.56, 25th percentile: 2898.08), *Gpr164⁻/⁻* mice (Max: 61819.4, Min: 33753.75, Median: 46478.5, 75th percentile: 57968.86, 25th percentile: 35610.43). *Fcgbp*; WT (Max: 10462.01, Min: 4673.64, Median: 6590.51, 75th percentile: 8771.93, 25th percentile: 5270.48), *Gpr164⁻/⁻* mice (Max: 42111.05, Min: 19869.32, Median: 30503.14, 75th percentile: 36509.74, 25th percentile: 24056.38). *Alpi*; WT (Max: 476.45, Min: 70.58, Median: 99.65, 75th percentile: 420.95, 25th percentile: 72.3), *Gpr164⁻/⁻* mice (Max: 735.96, Min: 36.91, Median: 50.65, 75th percentile: 669.55, 25th percentile: 42.53). *Slc26a2*; WT (Max: 9587.52, Min: 1469.87, Median: 8092.53, 75th percentile: 9096.25, 25th percentile: 1761.33), *Gpr164⁻/⁻* mice (Max: 2252.11, Min: 1554.71, Median: 1612.75, 75th percentile: 2115.93, 25th percentile: 1568.18). *Scl4a2*; WT (Max: 2617.93, Min: 1528.58, Median: 2202.13, 75th percentile: 2428.77, 25th percentile: 1806.74), *Gpr164⁻/⁻* mice (Max: 1744.72, Min: 1483.07, Median: 1682.66, 75th percentile: 1715.75, 25th percentile: 1528.81).
(C) Schematic diagram of Wnt/β-catenin signaling pathway (upper), and heatmap of representative genes, involved in Wnt/β-catenin signaling, with significantly differences between WT and *Gpr164⁻/⁻* mice (bottom) (n = 5). (D) Representative images of immunofluorescent staining for β-catenin. The cross-sections of colon obtained from WT and *Gpr164⁻/⁻* mice were stained with anti- β-catenin antibody (n = 5). Higher-magnification images of immunofluorescent staining for β-catenin is shown (right). DAPI was used for nuclei staining. Scale bar, 100 μm. Source data are available online for this figure.

hyperplasia (Figs. 2B–D and EV2A,B). However, there were no significant differences in p53 protein expression and activation of caspase-3 between genotypes (Figs. 1G and EV3B), indicating that *Gpr164* deletion results in increased cell proliferation but not in reduced apoptosis. In the present study, we showed that loss of *Gpr164* caused aberrant and ectopic expression of β-catenin protein (Fig. 4D), which induces overactivation of Wnt signaling. By using PNU-74654 (Wnt pathway inhibitor), we revealed that abnormal Wnt signaling in *Gpr164⁻/⁻* mice led to increased proliferation of intestinal epithelial cells and enhanced differentiation into goblet cells (Fig. 5A–F). Furthermore, PNU-74654 restored the expression of tight junction marker genes in *Gpr164⁻/⁻* mice (Fig. 5G), suggesting that colonic epithelial cells of *Gpr164⁻/⁻* mice may have defects in the formation of organized monolayer because of the abnormal Wnt signaling.

The mucus layer is important for a physical barrier against luminal pathogens in the colon. In this study, we observed strong staining for alcian blue-PAS in colon of *Gpr164⁻/⁻* mice (Fig. 3A), which is consistent with the results obtained by RNA-seq analysis (Fig. 4B). These results suggest that loss of GPR164 results in increased differentiation into goblet cells. However, thickness of mucus layer in *Gpr164⁻/⁻* mice was reduced as compared to WT mice (Fig. 3B). Considering that GPR164 plays an important role in regulating differentiation of colonic epithelial cells, this receptor may affect the function of goblet cells. Indeed, increased stored mucins were observed in goblet cells of *Gpr164⁻/⁻* mice (Fig. 3A), suggesting the possibility that the function of goblet cells such as mucus secretion is impaired in *Gpr164⁻/⁻* mice. The colon has a two-layered mucus, inner and outer mucus layer. The commensal bacteria colonize the outer mucus layer which provides the ideal habitat as a nutritional source and microbial adhesion sites. Thus, a defect in the outer mucus layer alters the microbial composition, leading to gut dysbiosis. The inner mucus layer serves as a physical barrier to prevent pathogenic infection. The inner mucus defects

allow bacteria to reach colonic epithelial cells, which causes gut inflammation by stimulating host immune response. Notably, our study revealed that loss of *Gpr164* resulted in gut microbial dysbiosis and inflammation (Fig. 3C–F). Therefore, our findings imply that GPR164 maintains mucus barrier function by regulating goblet cell maturation. Furthermore, Our 16S rRNA sequencing revealed that the presence of *Akkermansiaceae*, *Bifidobacteriaceae*, *Prevotellaceae* in the colon of *Gpr164⁻/⁻* mice tended to decrease (Fig. 3D). In particular, *Akermansiaceae*, mainly *Akermansia mucinifila* (*A. muciniphila*), is known to be associated with inflammatory diseases and cancer. *A. muciniphila* degrades mucin in the colon, which causes inflammation and the subsequent development of colorectal cancer (Grenda et al, 2022). Contrary to this, a marked increase in the abundance of *A. muciniphila* is observed in the patients treated with chemotherapy for colorectal cancer, and its abundance is positively correlated with the therapeutic effect (Grenda et al, 2022). These reports suggest that the abundance of *A. muciniphila* is strongly associated with colorectal cancer, and hence GPR164 might be involved in the development of colorectal cancer through affecting gut microbial composition. Additionally, changes in the luminal microenvironment caused by pathogen challenge and dietary manipulations affect the *Gpr164* gene expression (Priori et al, 2015). These findings, including ours, indicate a positive correlation between GPR164 and gut microbial diversity.

Butyrate is an important gut microbial metabolite for the maintenance of mucus barrier function through promoting expression and release of mucins (Willemsen et al, 2003; Jung et al, 2022). Considering our results shown in Fig. 3G, decreased level of butyrate may be insufficient to stimulate mucin secretion. Including a possibility that butyrate-mediated activation of GPR164 facilitates mucin secretion, further studies are required to understand the molecular mechanisms of how GPR164 regulates mucus barrier function. Besides the maintenance of mucosal barrier,

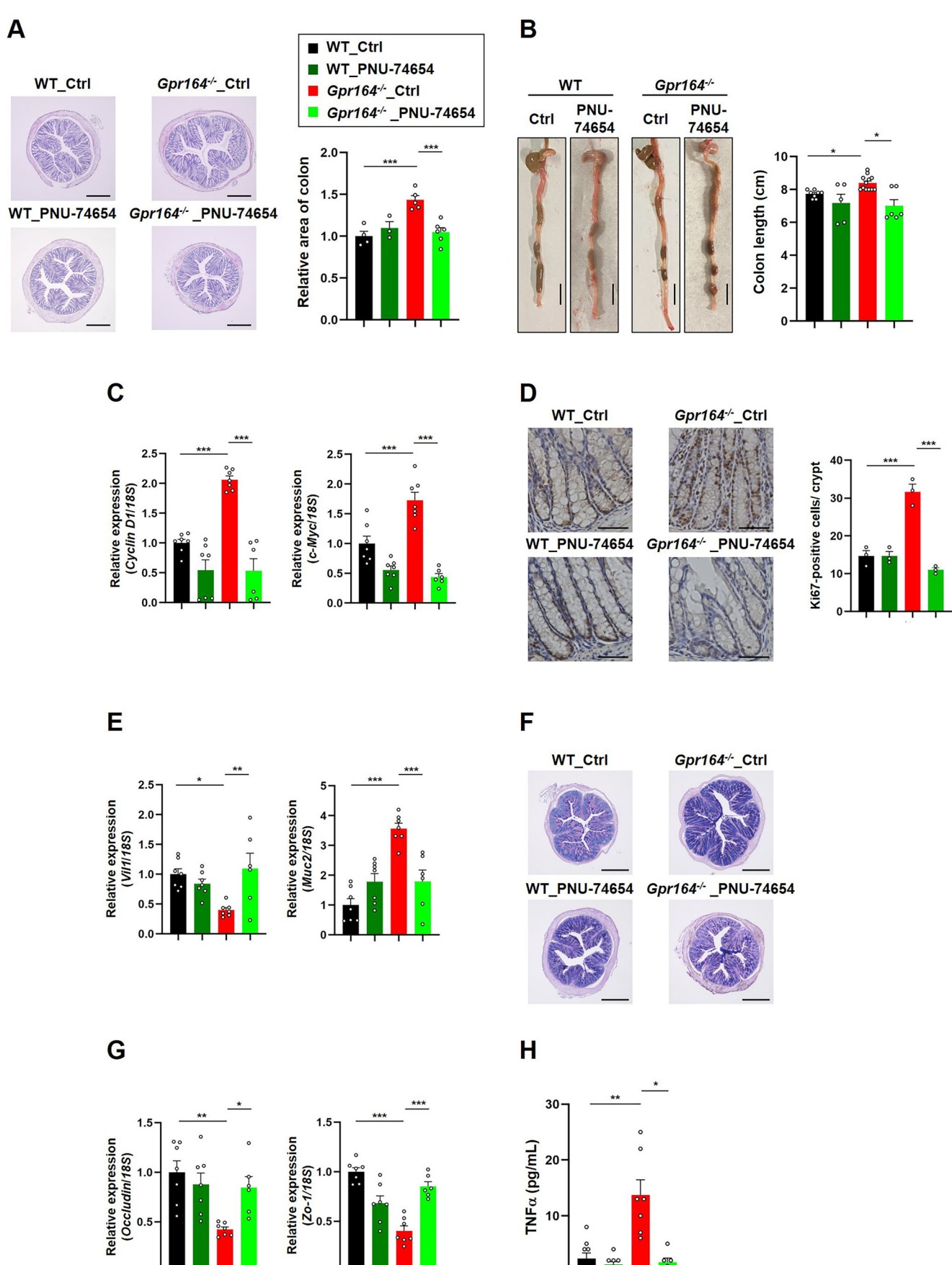

**Figure 5. PNU-74654 ameliorates colonic abnormalities in *Gpr164$^{-/-}$* mice.**

(A) Representative images of colon sections stained with hematoxylin-eosin (HE). WT and *Gpr164$^{-/-}$* mice were injected intraperitoneally with PNU-74654 (15 mg/kg body weight) every 2 days for 3 weeks. The cross-sections of colon obtained from WT and *Gpr164$^{-/-}$* mice were stained with HE, and the area of colon was measured using ImageJ software (*n* = 3–6). Scale bar, 500 μm. Error bars represent the mean ± SEM. ***$P$ = 0.0007; WT_Ctrl vs *Gpr164$^{-/-}$*_Ctrl, ***$P$ = 0.0008; *Gpr164$^{-/-}$*_Ctrl vs *Gpr164$^{-/-}$*_PNU-74654 (Tukey–Kramer test). (B) Representative images of colon (*n* = 5–12). Scale bar, 1 cm. Error bars represent the mean ± SEM. *$P$ = 0.012; WT_Ctrl vs *Gpr164$^{-/-}$*_Ctrl, **$P$ = 0.0018; *Gpr164$^{-/-}$*_Ctrl vs *Gpr164$^{-/-}$*_PNU-74654 (Tukey–Kramer test). (C) The mRNA expression levels of *Cyclin D1* and *c-Myc* determined by qRT-PCR. Total RNA was extracted from colon tissue of WT and *Gpr164$^{-/-}$* mice (*n* = 6–7). Error bars represent the mean ± SEM. ****$P$ < 0.0001; WT_Ctrl vs *Gpr164$^{-/-}$*_Ctrl of *Cyclin D1* gene, ****$P$ < 0.0001; *Gpr164$^{-/-}$*_Ctrl vs *Gpr164$^{-/-}$*_PNU-74654 of *Cyclin D1* gene, ***$P$ = 0.0003; WT_Ctrl vs *Gpr164$^{-/-}$*_Ctrl of *c-Myc* gene, ****$P$ < 0.0001; *Gpr164$^{-/-}$*_Ctrl vs *Gpr164$^{-/-}$*_PNU-74654 of *c-Myc* gene (Tukey–Kramer test). (D) Representative images of immunohistochemical staining for Ki67 in colon. The cross-sections of colon obtained from WT and *Gpr164$^{-/-}$* mice were stained with anti-Ki67 antibody. DAB was used for the detection of Ki67 protein, and hematoxylin was used for nuclei staining (*n* = 3). Scale bar, 100 μm. Error bars represent the mean ± SEM. ***$P$ = 0.0001; WT_Ctrl vs *Gpr164$^{-/-}$*_Ctrl, ****$P$ < 0.0001; *Gpr164$^{-/-}$*_Ctrl vs *Gpr164$^{-/-}$*_PNU-74654 (Tukey–Kramer test). (E) The mRNA expression levels of *Vil1* and *Muc2* determined by qRT-PCR. Total RNA was extracted from colon tissue of WT and *Gpr164$^{-/-}$* mice (*n* = 6–7). Error bars represent the mean ± SEM. *$P$ = 0.0145; WT_Ctrl vs *Gpr164$^{-/-}$*_Ctrl of *Vil1* gene, **$P$ = 0.0059; *Gpr164$^{-/-}$*_Ctrl vs *Gpr164$^{-/-}$*_PNU-74654 of *Vil1* gene, ****$P$ < 0.0001; WT_Ctrl vs *Gpr164$^{-/-}$*_Ctrl of *Muc2* gene, ***$P$ = 0.0006; *Gpr164$^{-}$*-_Ctrl vs *Gpr164$^{-/-}$*_PNU-74654 of *Muc2* gene (Tukey–Kramer test). (F) Representative images of colon sections stained with alcian blue-PAS (*n* = 3). Scale bar, 500 μm. (G) The mRNA expression levels of *Occludin* and *Zo-1* determined by qRT-PCR. Total RNA was extracted from colon tissue of WT and *Gpr164$^{-/-}$* mice (*n* = 6–7). Error bars represent the mean ± SEM. **$P$ = 0.0018; WT_Ctrl vs *Gpr164$^{-/-}$*_Ctrl of *Occludin* gene, *$P$ = 0.0338; *Gpr164$^{-/-}$*_Ctrl vs *Gpr164$^{-/-}$*_PNU-74654 of *Occludin* gene, ****$P$ < 0.0001; WT_Ctrl vs *Gpr164$^{-/-}$*_Ctrl of *Zo-1* gene, ****$P$ < 0.0001; *Gpr164$^{-/-}$*_Ctrl vs *Gpr164$^{-/-}$*_PNU-74654 of *Zo-1* gene (Tukey–Kramer test). (H) Plasma levels of TNFα (*n* = 6–8). Error bars represent the mean ± SEM. **$P$ = 0.0071; WT_Ctrl vs *Gpr164$^{-/-}$*, *$P$ = 0.018; *Gpr164$^{-/-}$*_Ctrl vs *Gpr164$^{-/-}$*_PNU-74654 (Dunn's test). Source data are available online for this figure.

butyrate regulates the tight junction barrier by an increase in the expression of tight junction proteins (Ikeda et al, 2022). However, the molecular mechanisms underlying butyrate-mediated enhancement of tight junction barrier remain elusive. Here, we show that butyrate enhances the intestinal barrier function in part through GPR164 activation (Fig. 6C–E). Our findings thus provide a novel insight into how butyrate regulates the intestinal barrier function. In addition, butyrate serves as an immunomodulator with anti-inflammatory effects in the intestinal epithelial cells. Butyrate suppresses pro-inflammatory cytokine production, and ameliorates the development of colitis by promoting differentiation of regulatory T cells[28,29]. In *Gpr164$^{-/-}$* mice, elevated plasma TNFα levels were observed (Fig. 3F). Given that inflammation and dysbiosis contribute to epithelial injury, the colonic hyperplasia in *Gpr164$^{-/-}$* mice may be caused by the increase in colonic epithelial renewal. During inflammation, Wnt signaling promotes the intestinal stem cell (ISC) self-renewal, and Hippo signaling prevents ISCs from excessive proliferation leading to tumorigenesis through dampening Wnt signaling. Therefore, these two key factors for controlling ISC self-renewal are critical to the regulation of epithelial proliferation. Further studies are needed to evaluate the molecular links between ISC self-renewal and GPR164 during injury (Totaro et al, 2018).

Both intestinal barrier dysfunction and gut microbial dysbiosis are the characteristic features of IBD, and the chronic inflammation in IBD patients increases a risk of developing colorectal carcinoma (CRC) (Brackmann et al, 2009). Overactivation of Wnt signaling is also known as a hallmark of CRC. Intriguingly, GPR164 is overexpressed in human primary prostate cancers and used as a biomarker of human prostate cancer (Weng et al, 2006). As well as in the prostate cancer, high expression of GPR164 is observed in lung carcinoids and digestive neuroendocrine carcinomas (Giandomenico et al, 2013; Cui et al, 2013). Notably, it has been reported that nonanoate and decnoate, identified as GPR164 agonist, suppresses proliferation of prostate cancer cell line LNCaP (Miyamoto et al, 2019). Therefore, these facts provide a strong correlation GPR164 and cancers. Overall, this study uncovers the physiological functions of GPR164, which will help us understand the relation between diet and diseases.

## Methods

**Reagents and tools table**

| Reagent/resource | Reference or source | Identifier or catalog number |
|---|---|---|
| **Experimental models** | | |
| C57BL/6JJmsSlc (*M. musculus*) | Japan SLC | |
| GPR164$^{-/-}$ (*M. musculus*) | This study | Table EV1 |
| **Recombinant DNA** | | |
| pcDNA3.1_HA-*Or51e1* | This study | Materials and methods |
| pcDNA3.1_*Rtp1S* | This study | Materials and methods |
| pcDNA3.1_*Rtp4* | This study | Materials and methods |
| peSpCAS9(1.1)-2xsgRNA | Addgene | #80768 |
| eSpCAS9(1.1)-2xsgRNA and pDonor-tBFPNLS-Neo | Addgene | #80766 |
| **Antibodies** | | |
| Anti-p53 antibody | Santa Cruz | sc-126 |
| Anti-tubulin antibody | Sigma | T5168 |
| Anti-β-catenin | Proteintech | 51067-2-AP |
| Anti-Ki67 antibody | Proteintech | 28074-1-AP |
| **Oligonucleotides and other sequence-based reagents** | | |
| Genotyping primers | This study | Fig. EV1 |
| Cloning primers | This study | Materials and methods |
| qPCR primers | This study | Table EV1 |
| 16S rRNA sequencing primer | This study | Materials and methods |
| Bacterial primers | This study | Materials and methods |

| Reagent/resource | Reference or source | Identifier or catalog number |
|---|---|---|
| **Chemicals, enzymes and other reagents** | | |
| CE-2 | Japan CLEA | 30088828 |
| HFD | Research diet | D12492 |
| FITC-dextran | Sigma | 60842-46-8 |
| DMEM | Wako | 043-30085 |
| FBS | Cosmo Bio | S-FBS-CO-015 |
| PenStrept | Gibco | 15140-122 |
| Lipofectamine 2000 | Invitrogen | 11668019 |
| G418 | Wako | 074-06801 |
| Culture insert | Falcon | 353095 |
| Sodium butyrate | Tokyo Chemical Industry | 156-54-7 |
| Sodium palmitate | Tokyo Chemical Industry | 408-35-5 |
| RNAiso plus | TAKARA | 9109 |
| Moloney murine leukemia virus reverse transcriptase | Invitrogen | 28025013 |
| 3-isobutyl 1-methylxantine | Sigma | 28822-58-4 |
| cAMP EIA kit | Cayman Chemical | 581001 |
| Folskolin | Sigma | 66575-29-9 |
| Crystal violet | Wako | 031-04852 |
| Alexa Fluor 488 | Invitrogen | A11008 |
| Goat anti-rabbit IgG antibody (H + L), peroxidase | Vector Laboratories | PI-1000 |
| Diaminobenzidine | DOJINDO | 349-00903 |
| Sodium citrate buffer | KANTO KAGAKU | 73114 |
| Carnoy's solution | Wako | 034-17711 |
| Alcian blue-PAS stain kit | Scy Tek | APS-1 |
| FastDNA SPIN kit | MP Biomedicals | 116560200-CF |
| AMPure XP | Beckman Coulter | A63881 |
| RNeasy mini kit | Qiagen | 74904 |
| RNA 6000 Nano kit | Agilent Technologies | 5067-1511 |
| NEBNext® Ultra™ II Directional RNA Library Prep Kit | Illumina | E7760S |
| NEBNext Multiplex Oligos for Illumina | Illumina | E7335S |
| IntestiCult organoid growth medium | Veritas | ST-06010 |
| **Software** | | |
| CRISPOR | http://crispor.tefor.net | |
| trimmomatic (v0.39) | https://github.com/usadellab/Trimmomatic/releases/download/v0.39/Trimmomatic-0.39.zip | |
| FastQC (v0.11.8.-2) | https://sourceforge.net/projects/fastqc.mirror/files/v0.11.8/ | |
| STAR (v2.7.10a) | https://github.com/alexdobin/STAR/releases/download/2.7.11b/STAR_2.7.11b.zip | |
| GraphPad Prism | https://www.graphpad.com/ | |
| Primer 3 | https://primer3.ut.ee/ | |
| **Other** | | |
| Millicell-ERS system | Millipore | MERS00002 |
| StepOne real-time PCR system | Applied Biosystems | |
| SYBR Premix Ex Taq II | TAKARA | RR820A |
| Preparation of cross-sections | Kyoto Institute of Nutrition & Pathology, Inc | |
| GCMS-QP2010 Ultra system | Shimazu | |
| MiSeq platform | Illumina | |
| Agilent 2100 Bioanalyzer | Agilent Technologies | |
| NovaSeq 6000 platform | Illumina | |
| Kyoto Encyclopedia of Genes and Genomes (KEGG) database | http://www.genome.jp/kegg/ | |
| Fluorescence microscope | Keyence | BZ-X710 |

## Animals

Male C57BL/6J mice were purchased from Japan SLC and *Or51e1* gene knockout (*Gpr164$^{-/-}$*) mice were generated by using the CRISPR/Cas9 system in wild-type C57BL/6J zygotes (Fig. EV1D–F). *Gpr164$^{-/-}$* mice are viable and have no obvious abnormalities in behavior and fertility, and mice which were obviously injured by fighting were excluded from the study. C57BL/6 J and *Gpr164$^{-/--}$* mice were housed at a temperature of 24 °C and 50% relative humidity under a 12 h light/dark cycle. For high-fat diet (HFD) feeding study, 6-week-old male mice were fed with either a normal chow diet (ND) (CE-2; CLEA Japan) or a HFD (D12492, 60% kcal fat; Research diet) for 10 weeks. For the assessment of intestinal permeability, mice withdrawn from food and water for 4 h were administrated a freshly prepared FITC-dextran (4 kDa; Sigma) at 0.6 g/kg body weight by oral gavage. After 3 h, blood was collected from the inferior vena cava, and plasma was separated by immediate centrifugation at 7000×*g* for 5 min at 4 °C. The fluorescent intensity of FITC-dextran within plasma samples was measured by FlexStation3 fluorescence plate-reader (emission 520 nm and excitation 490 nm). A standard curve was generated by serial dilution of FITC-dextran, and used for determining concentration of FITC-dextran in plasma samples. All animal experiments were performed in accordance with the guidelines of the Committee in the Ethics of Animal Experiments of Kyoto University Animal Experimental Committee (Lif-K25002). All efforts were made to minimize suffering. Sample randomization was done by a simple random sample.

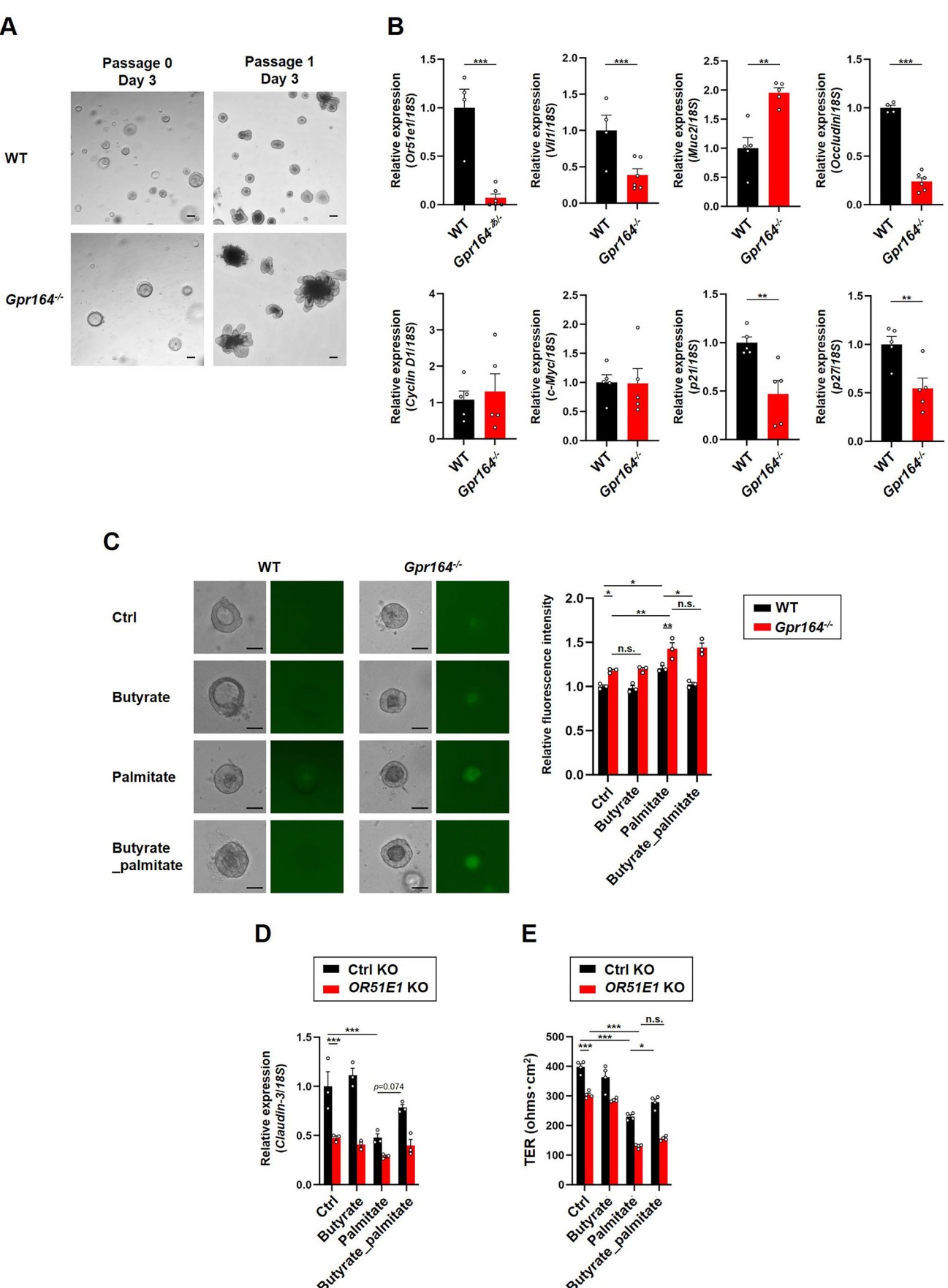

◀ **Figure 6.  Butyrate-mediated enhancement of epithelial barrier function is lost in *OR51E1*-knockout Caco-2 cells *and Gpr164−/−* mice.**

(A) Representative images of colonic organoids isolated from WT and *Gpr164−/−* mice ($n = 3$). Scale bar, 100 μm. (B) The mRNA expressions in colonic organoids determined by qRT-PCR. Total RNA was extracted from colonic organoids isolated from WT and *Gpr164−/−* mice ($n = 5$). Error bars represent the mean ± SEM. ****$P < 0.0001$; *Or51e1* gene, ***$P = 0.0001$; *Vil1* gene, **$P = 0.0015$; *Muc2* gene, ****$P < 0.0001$; *Occludin* gene, **$P = 0.0077$; *p21* gene, **$P = 0.01$; *p27* gene (Student's *t*-test). (C) Representative images of fluorescence in colonic organoids. Colonic organoids isolated from WT and *Gpr164−/−* mice were preincubated for 24 h in the presence or absence of 1 mM sodium butyrate, and followed by stimulation with or without 500 μM sodium palmitate for 24 h. Leakage of FITC-dextran into the lumen was assessed by fluorescence ($n = 3$). Scale bar, 50 μm. Error bars represent the mean ± SEM. *$P = 0.0406$; WT_Ctrl vs *Gpr164−/−*_Ctrl, *$P = 0.0131$; WT_Ctrl vs WT_Palmitate, **$P = 0.0028$; *Gpr164−/−*_Ctrl vs *Gpr164−/−*_Palmitate, *$P = 0.0348$; WT_ Palmitate vs WT_Butyrate/Palmitate, **$P = 0.0089$; WT_ Palmitate vs *Gpr164−/−*_Palmitate (Tukey–Kramer test). (D, E) *Claudin-3* mRNA expression and TER measurements. Control and *OR51E1* KO Caco-2 cells were preincubated for 24 h in the presence or absence of 1 mM sodium butyrate, and followed by stimulation with or without 500 μM sodium palmitate for 24 h. *Claudin-3* mRNA expression was determined by qRT-PCR (D, $n = 3$), and the effect of butyrate on epithelial barrier was determined by measuring the TER (E, $n = 4$). Error bars represent the mean ± SEM. ***$P = 0.0009$; Ctrl KO_Ctrl vs *OR51E1* KO_Ctrl, ***$P = 0.0009$; Ctrl KO_Ctrl vs Ctrl KO_Palmitate, $P = 0.074$; Ctrl KO_Palmitate vs Ctrl KO_Butyrate/Palmitate (D), ****$P < 0.0001$; Ctrl KO_Ctrl vs *OR51E1* KO_Ctrl, ****$P < 0.0001$; Ctrl KO_Ctrl vs Ctrl KO_Palmitate, ****$P < 0.0001$; *OR51E1* KO_Ctrl vs *OR51E1* KO_Palmitate, *$P = 0.0297$; Ctrl KO_Palmitate vs Ctrl KO_Butyrate/Palmitate (E) (Tukey–Kramer test). Source data are available online for this figure.

## Cell culture

Caco-2 and HEK293 cells were maintained in Dulbecco's modified Eagle's medium (DMEM) supplemented with 10% fetal calf serum, penicillin and streptomycin at 37 °C in 5% $CO_2$. To generate Or51e1-overexpressing cells, HEK293 cells were co-transfected with pcDNA3.1_HA-*Or51e1* and pcDNA3.1_*Rtp1S* (or *RTP4*) using lipofectamine 2000 (Invitrogen). *OR51E1*-deficient (*OR51E1* KO) Caco-2 cells were generated using the CRISPR/Cas9 system. Single guide RNA (sgRNA) targeting *OR51E1* (5'-atccgggt-caatgtcgtcta-3') was designed by using the online software CRISPOR (http://crispor.tefor.net). The sgRNA was cloned into peSpCAS9(1.1)-2xsgRNA vector (Addgene, #80768), and the recombinant peSpCAS9(1.1)-2xsgRNA and pDonor-tBFPNLS-Neo (Addgene, #80766) were co-transfected into Caco-2 cells using Lipofectamine 2000 (Invitrogen). For the selection of *OR51E1* KO cells, transfected cells were maintained in medium containing 250 mg/mL G418 (FUJIFILM Wako), and the colony grown from a single cell was isolated. Control KO cells were obtained by transfection with Cas9 expression vector alone. To assess the intestinal barrier function, Caco-2 cells were seeded on a cell culture insert (0.4 μm pore; Falcon) and transepithelial electrical resistance (TER) value was measured using Millicell-ERS system (Millipore). To evaluate the effect of butyrate on barrier function, Caco-2 cells seeded on a cell culture insert were preincubated with or without 1 mM sodium butyrate for 24 h, and stimulated with either 500 μM sodium palmitate or control DMSO. After 24 h stimulation, TER value was measured.

## Quantitative RT-PCR

Total RNA was extracted using an RNAiso Plus reagent (TAKARA), and cDNA was synthesized by using Moloney murine leukemia virus reverse transcriptase (Invitrogen). Quantitative PCR analysis was performed using the StepOne real-time PCR system (Applied Biosystems) with SYBR Premix Ex Taq II (TAKARA). Each value was normalized to *18S* rRNA and calculated using the $2^{-\Delta\Delta Ct}$ method. The primer sequences are shown in Table EV1.

## cAMP measurement

HEK293 cells transfected with either empty or Or51e1 expression vector were maintained in serum-free DMEM containing 500 μM of 3-isobutyl 1-methylxantine (Sigma) for 30 min. Cell were then stimulated with individual FFAs at 1 mM concentration for 10 min. The cAMP levels were determined using cAMP EIA kit (Cayman Chemical) according to the manufacturer's instructions. Samples treated with 2 μM folskolin (Sigma) were used as positive controls.

## Crystal violet staining

Caco-2 cells were seeded in 24-well plate ($1 \times 10^5$ cells/well) and fixed with 4% paraformaldehyde at the indicated times. Cells were then stained with 0.1% crystal violet (FUJIFILM Wako) for 15 min, and washed with PBS three times. The dye was dissolved in 95% ethanol, and absorbance was measured at $OD_{590}$ nm.

## Immunoblot and immunofluorescence analysis

Immunoblotting was performed according to a standard protocol using anti-p53 antibody (Santa Cruz, sc-126) and anti-tubulin antibody (Sigma, T5168). α-Tubulin was used as a loading control. For immunofluorescence analysis, 10% formalin-fixed colon sample was embedded in paraffin and cross-sectioned at 4 μm thickness (Kyoto Institute of Nutrition & Pathology, Inc). The cross-sections were labeled with either anti-β-catenin primary antibody (Protein-tech, 51067-2-AP), anti-cleaved caspase-3 (Cell signaling, 9661 T) and anti-CD4 (eBioscience, RM4-5) followed by staining with Alexa Fluor 488, and observed with a fluorescence microscope (Keyence, BZ-X710). DAPI was used for nuclei staining.

## HE staining and immunohistochemistry

Hematoxylin and eosin (HE) staining was performed by a standard method. The paraffin-embedded cross-section was deparaffinized and stained with eosin and hematoxylin. Immunohistochemistry was performed according to a standard protocol. Briefly, the paraffin-embedded cross-section was deparaffinized, and incubated with blocking serum. The cross-section was heated in 10 mM sodium citrate buffer (pH 6.0) for 50 min to retrieve antigen. The anti-Ki67 antibody (Proteintech, 28074-1-AP) was used as a primary antibody, and peroxidase-conjugated secondary antibody (Vector Laboratories, PI-1000) was used. The cross-section was visualized with diamino-benzidine (DAB) and counterstained with hematoxylin. For BrdU immunohistochemistry, the 8-week-old male C57BL/6 J and *Gpr164−/−* mice were injected with BrdU (100 mg/ kg body weight)

intraperitoneally and sacrificed 24 h later. The 10% formalin-fixed colon samples were embedded in paraffin and cross-sectioned at 4 µm thickness (Kyoto Institute of Nutrition & Pathology, Inc). For the detection of BrdU-incorporated cells, the paraffin-embedded cross-section was deparaffinized and treated with 2 M HCl for 30 min. The cross-section was then neutralized by incubating in 0.1 M sodium borate (pH 8.5) for 10 min. After washing with PBS, endogenous peroxidase activity was blocked by 0.3% hydrogen peroxide and blocking serum was added. The anti-BrdU antibody (Proteintech, 66241-1-IG) was used as a primary antibody, and peroxidase-conjugated secondary antibody (Vector Laboratories, PI-2000-1) was used. The cross-section was visualized with diaminobenzidine (DAB) and counterstained with hematoxylin.

## Alcian blue-periodic acid Schiff (PAS) staining

The colon sample was fixed in Carnoy's solution (FUJIFILM Wako) and embedded in paraffin. Alcian blue-PAS staining of mucins was performed using alcian blue-PAS stain kit (Scy Tek), following the manufacturer's instructions. In short, deparaffinized colon section was treated with 3% acetic acid for 2 min, followed by staining with alcian blue (pH 2.5) for 15 min. After washing with running tap water, the tissue section was treated with periodic acid for 5 min. The tissue section was then incubated with Schiff's solution, and stained with hematoxylin. The cross-sections were obtained from six independent individuals, and relative thickness of mucin layer was measured using ImageJ software.

## Measurement of SCFAs

SCFAs in the colon contents were determined as described previously (Miyamoto et al, 2019). In brief, the samples containing internal control (2-ethyl butyrate) were mixed with diethyl ether and centrifuged at 3000×$g$ for 5 min. SCFA-containing ether layers were collected and subjected to gas chromatography-mass spectrometry using a GCMS-QP2010 Ultra system (Shimadzu). Using the calibration curves for SCFAs, SCFA concentration in each sample was determined.

## Lactulose/mannitol test

Lactulose/mannitol test was performed according to the previous study (Baraga Neto et al, 2010). The 10-week-old male C57BL/6 J and $Gpr164^{-/-}$ mice were fasted overnight, and 5 mg of lactulose (FUJIFILM Wako) and 12.5 mg of mannitol (FUJIFILM Wako) were administrated to mice by oral gavage. The urine was collected within the next 24 h, and the urinary lactulose and mannitol were determined using EnzyChrom™ D-mannitol Assay Kit (BioAssay Systems) and Lactulose Assay Kit (Megazyme), respectively. The excretion ratio of lactulose to mannitol was used to evaluate the intestinal permeability.

## 16S rRNA sequencing of gut microbes

Fecal DNA was extracted using FastDNA SPIN kit for feces (MP Biomedicals), according to the manufacturer's instructions. To determine the microbial taxa, the V3-V4 regions of the bacterial 16S rRNA gene were amplified using following primers; (forward) 5'-TCGTCGGCAGCGTCAGATGTGTATAAGAGACAGCCTAC GGGNGG CWGCAG-3', (reverse) 5'-GTCTCGTGGGCTCGGA GATGTGTATAAGAGACAGGA CTACHVGGGTATCTAATCC-3'. Amplified products from each sample were purified by AMPure XP (Beckman Coulter) and sequenced using the MiSeq platform (Illumina). Raw data were processed using quantitative insights into microbial ecology (QIIME) pipeline, and analyzed by the MiSeq reporter software with the SILVA database (Illumina). The diversity of gut microbiota was determined by the QIIME script core_diversity_analyses.py. The statistical significance of difference between groups was assessed by a permutational multivariate analysis of variance (QIIME script compare_categories.py). The abundance of Firmicutes and Bacteroides in the gut microbiota was quantified by qRT-PCR. Standard curves for quantification comprised a series of ten-fold serial dilutions in the range of $10^8$ to $10^0$ copies of target 16S rRNA genes from each specific strains. Bacterial primer sequences are as follows; Firmicutes, 5'-GGA-GYATGTGGTTTAATTCGAAGCA-3' (forward) and 5'-AGCT-GACGACAACCATGCAC-3' (Reverse); Bacteroides, 5'-CRAAC AGGATTAGATACCCT-3' (Forward) and 5'-GGTAAGGTTCC TCGCGTAT-3' (Reverse).

## RNA sequencing

Total RNA was extracted from the colon of WT and $GPR164^{-/-}$ mice using an RNAiso Plus reagent (TAKARA) and RNeasy mini kit (Qiagen). The quality of RNA samples was determined by Agilent 2100 Bioanalyzer with an RNA 6000 Nano kit (Agilent Technologies). RNA sequencing libraries were created using the NEBNext® Ultra™ II Directional RNA Library Prep Kit (Illumina) and NEBNext Multiplex Oligos for Illumina (Dual Index Primers Set 1). Paired-end sequencing was performed using an Illumina NovaSeq 6000 (150 bp read lengths; ~4 Gb). The data obtained from RNA sequencing were trimmed with trimmomatic (v0.39) software to eliminate adapters and inadequate quality bases (Bolger et al, 2014), and FastQC (v0.11.8.-2) software was used for quality control of the trimmed sequence. The reads were aligned to the mouse reference genome (NCBI GRCm39) using STAR software (v2.7.10a). To obtain differentially expressed genes (DEGs) across all comparisons, the raw read counts were subjected to relative log expression normalization. Based on the following criteria, DEGs were determined; a false discovery rate (FDR)-adjusted $p$ value <0.05 (Benjamini–Hochberg procedure) and an absolute log2 fold change >0.5. Gene set enrichment analysis was performed using the Kyoto Encyclopedia of Genes and Genomes (KEGG) database (http://www.genome.jp/kegg/).

## Organoid culture

Mouse colonic segments of WT and $GPR164^{-/-}$ mice were collected in a PBS solution containing 5 mM EDTA, and rotated for 40 min at 4 °C. The colonic crypts were isolated from the segments by repetitive pipetting, and filtered through a cell strainer. The isolated crypts were embedded in Matrigel, and maintained in human IntestiCult organoid growth medium (Veritas) containing penicillin and streptomycin. Organoids were maintained at 37 °C in 5% $CO_2$, and the medium was changed every other day. The assessment of barrier function using organoids was performed as described previously (Hu et al, 2023). Briefly, the colonic organoids were preincubated with or without 1 mM sodium butyrate for 24 h, and stimulated with either 500 µM sodium palmitate or control

DMSO for 24 h. After stimulation, the organoids were washed with PBS, and incubated with FITC-dextran (4 kDa; Sigma) at 1.25 μM concentration for 1 h. To remove FITC-dextran from the media, organoids were gently washed with PBS, and fluorescence within the organoids was observed with a fluorescence microscope (Keyence, BZ-X710). The fluorescence intensity within the organoids isolated from three independent individuals was measured using ImageJ software.

## Statistical analysis

All values are shown as mean ± standard error of the mean (SEM). Statistical analyses were performed using GraphPad Prism software (GraphPad Software). The Shapiro–Wilk test was used for the assessment of data normality. The statistical significance of differences between two groups was determined by a two-tailed unpaired Student's *t*-test, while that of differences among multiple groups was determined by one-way ANOVA followed by the Turkey–Krammer test or Dunn's test. Time-dependent changes in cell growth and TER values were analyzed by using a two-way ANOVA followed by the Sidak's post hoc multiple comparison test. Statistical significance was defined as $P < 0.05$.

## Data availability

The source data shown in RNA-seq analysis and 16S rRNA sequence have been deposited into the DNA Data Bank of Japan (DDBJ) under the accession no. E-GEAD-1039 and DRA020128 (https://www.ncbi.nlm.nih.gov/sra/?term=DRA020128), respectively.

The source data of this paper are collected in the following database record: biostudies:S-SCDT-10_1038-S44319-025-00611-5.

## Peer review information

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

## Acknowledgements

This study was supported by research grants from the JSPS KAKENHI (22K21208, 23K16794, and 25K21072 to TI), Takeda Science Foundation (to TI), AMED (JP23gm1510011 to IK), JSPS KAKENHI (JP25H01097 to IK) and JST-MOONSHOT (JPMJMS2023 to IK). This work was carried out by the joint research program of the Institute for Molecular and Cellular Regulation, Gunma University.

## Author contributions

**Takako Ikeda**: Conceptualization; Data curation; Formal analysis; Supervision; Funding acquisition; Validation; Investigation; Visualization; Writing—original draft; Project administration; Writing—review and editing. **Yuki Masujima**: Investigation. **Keita Watanabe**: Investigation. **Akari Nishida**: Investigation. **Mayu Yamano**: Investigation. **Miki Igarashi**: Investigation. **Nobuo Sasaki**: Methodology. **Hironori Katoh**: Methodology. **Ikuo Kimura**: Supervision; Funding acquisition.

Source data underlying figure panels in this paper may have individual authorship assigned. Where available, figure panel/source data authorship is listed in the following database record: biostudies:S-SCDT-10_1038-S44319-025-00611-5.

## Disclosure and competing interests statement

The authors declare no competing interests.

# Expanded View Figures

**Figure EV1.  Generation of Or51e1-overexpressing cells, *OR51E1*-knockout cells, and *Or51e1*-knockout (*Gpr164*⁻ᐟ⁻) mice.**

(A) Representative images of immunofluorescent staining for mouse Or51e1 in HEK293 cells. Cells were co-transfected with HA-tagged Or51e1 and receptor-transporting protein (left; RTP1S, right; RTP4), and stained with anti-HA antibody (green). DAPI was used for nuclei staining. Scale bar, 50 μm. (B) *OR51E1* expression in Caco-2 cells. *OR51E1*-deficient (*OR51E1* KO) cells were generated using the CRISPR/Cas9 system. For the detection of *OR51E1* expression, PCR amplification was done with the indicated primers, and the PCR products were separated on 1% agarose gels (left). The *OR51E1* mRNA expression level was determined by qRT-PCR ($n = 4$) (right). Error bars represent the mean ± SEM. **$P = 0.0021$ (Student's $t$-test). (C) Representative image of p53 protein expression. Cell extracts from Control KO or *OR51E1* KO cells were subjected to immunoblot analysis using an anti-p53 or anti-αTubulin antibody ($n = 3$). (D) Schematic representation of CRISPR/Cas9 targeting sites in *Or51e1* gene. *Or51e1* gene knockout (*Gpr164*⁻ᐟ⁻) mice were generated by using the CRISPR/Cas9 system in wild-type C57BL/6J zygotes. Bold letters indicate the coding region of *Or51e1* gene. Red or Green letters indicate guide RNA (gRNA) and protospacer adjacent motif (PAM), respectively. (E) For the detection of the wild-type and mutant alleles, PCR amplification was done with the indicated primers, and the PCR products were separated on 1% agarose gels. (F) The *Or51e1* mRNA expression level in colon was determined by qRT-PCR ($n = 3$). Error bars represent the mean ± SEM. **$P = 0.0025$ (Student's $t$-test). Source data are available online for this figure.

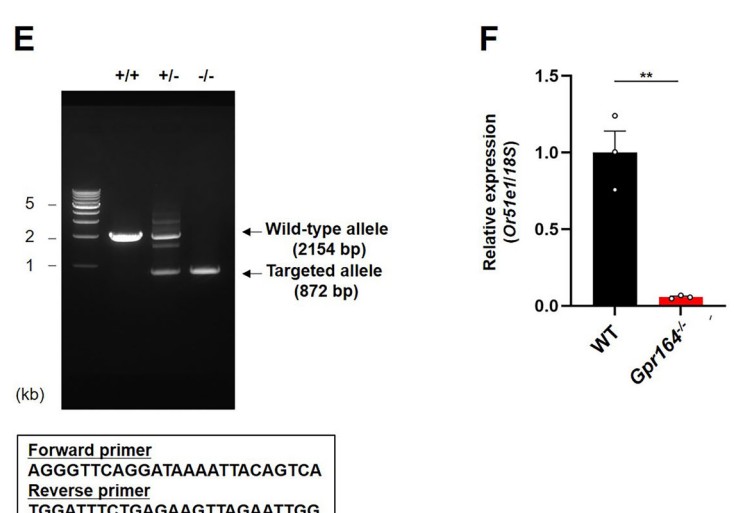

**A**

pcDNA3.1-*Or51e1*
_pcDNA3.1-*Rtp1S*

pcDNA3.1-*Or51e1*
_pcDNA3.1-*Rtp4*

**B**

Ctrl KO    *OR51E1* KO

1000
500
300
100

(bp)

← 253 bp

Forward primer
GCTATGTGGCCATCTGTCAC
Reverse primer
GCGGAGATGATGACGATAAG

Relative expression
(*OR51E1/18S*)

2.0
1.5
1.0
0.5
0.0

**

Ctrl KO    OR51E1 KO

**C**

Ctrl KO    *OR51E1* KO

250
150
100
75
50
37

(kDa)

← p53
(53 kDa)

Ctrl KO    *OR51E1* KO

250
150
100
75
50
37

(kDa)

← αTublin
(50-55 kDa)

**D**

*Or51e1*

1 kb

CTCTTGATTAAATAATGGAAGTTAAGACTT<u>AGGGTTCAGGATAAAATTACAGTCA</u>
Forward primer
TGTTTTTATTGAAATAGGCTTGATCATAATAAAATAGATGTTATTAAATTAATATTGA
AATGGCTTCTATTCCTGTGTGACATTAAAGTTATGATTAGGGCTAAATTAGATCAT
TAGATTATTGAGCAAGTTATATTTCAATGTCAGTTTATGTTATAACTATCACTGATG
GACTTACCTGATAATCACCTTAGGATGGACTGATGAAATGGTCATGATTACAGAA
TTAAAATTTTTTCATTTAATATTTCTCCATCCATAGTTTCATTTTCCAGAATCTGAC
AATAACTTGGATGGTTAATATAATCAATATTAAACATAGGAAGTATTTGATTTTTAG
TTTGAGAAACTAAATGATTAATAATATGAGAATGTTATCTTATTATTTCACTTAATAT
ATAGGTAGGATTAACAAGTAGAACTAAGTGGATTTTTCATCTGGTACTTGATTTAT
GGCTAGATAAAATTTAGGGTAGGCTCAGAGTTTGGATAGGCAATGGAACTAGAG
TTTTGTTAGAGAATTGGA<u style="color:red">CACTAGGATCCAGGGTGGTA</u>TGGCATTGAAATTCTA
*Or51e1* gRNA1_PAM
GAGGACGTGACAATGACTTGTCTTTGTATTTCAGCTTCAGTCTTCCTGGTACTG
GCTACATCCTGATTCCTTCAGT**ATGGTGGGCTTCAATAGCAATGAATCCAGTG**

**ATGTGCCTTTCATTGGATTGTCGATGGTGCACCGTTTCAGCAAGAGGTCCAG
GCGTGACTCTCTCCTGCCTGTCATTATGGCTAACATCTATCTGCTAGTTCCTCC
TGTGCTCAACCCCATTGTCTATGGAGTAAAGACAAAAGTGGAGATCCGGCAG
CGCATTCTTCGTCTTTTCCTCGTGACCACACACACTTCAGATCACTAG**GAAAT
CATGATGAAACCTTCCTCCATTCATTTAAGTTCTGTAACTCACACTTTAGTATGAT
ATCTTGGAAGACAGTATTAAGAAAATAAATCTTAATAAAAATATAGCTCAGATCTT
TCAAGGATGAAA<u style="color:red">CTTGCTGTGGAAATTCTACG</u>TGGAATGGAAGAACACCTGCA
*Or51e1* gRNA2_PAM
ATCTCCATTTTCTAATATTACATTCATTCTTTTGTTTTTTTCTCTAGATAATTATTAA
GATCAAGACTTGTGTTTTGAAAGTTATTTCTCACAATTTTATCACTACTTCCAAAT
TTCAATTCATTTCACTGATGATCGTTCACAGCATTTGGAGATGGAGCAACACATC
CAAAATGTTCATCAAAGGCATAACAAAAGAAAAATAAACACAAAACTATAATAAAA
TGATATTATCTCAATTTAAAACCTCATTTCCTCATCAGAATTCACAATGACTT<u>TGGAT
TTCTGAGAAGTTAGAATTGG</u>ATTCCTTATTCAAAACAATCTCCAAAGAAATATTGA
Reverse primer
TTCCTTATTCAAAACAATCTCCAAAGAAATATTGGTTTTCTTCTGGGTCCCAGGT

**E**

+/+    +/-    -/-

5
2
1

(kb)

← Wild-type allele
(2154 bp)
← Targeted allele
(872 bp)

Forward primer
AGGGTTCAGGATAAAATTACAGTCA
Reverse primer
TGGATTTCTGAGAAGTTAGAATTGG

**F**

Relative expression
(*Or51e1/18S*)

1.5
1.0
0.5
0.0

**

WT    *Gpr164*−/−

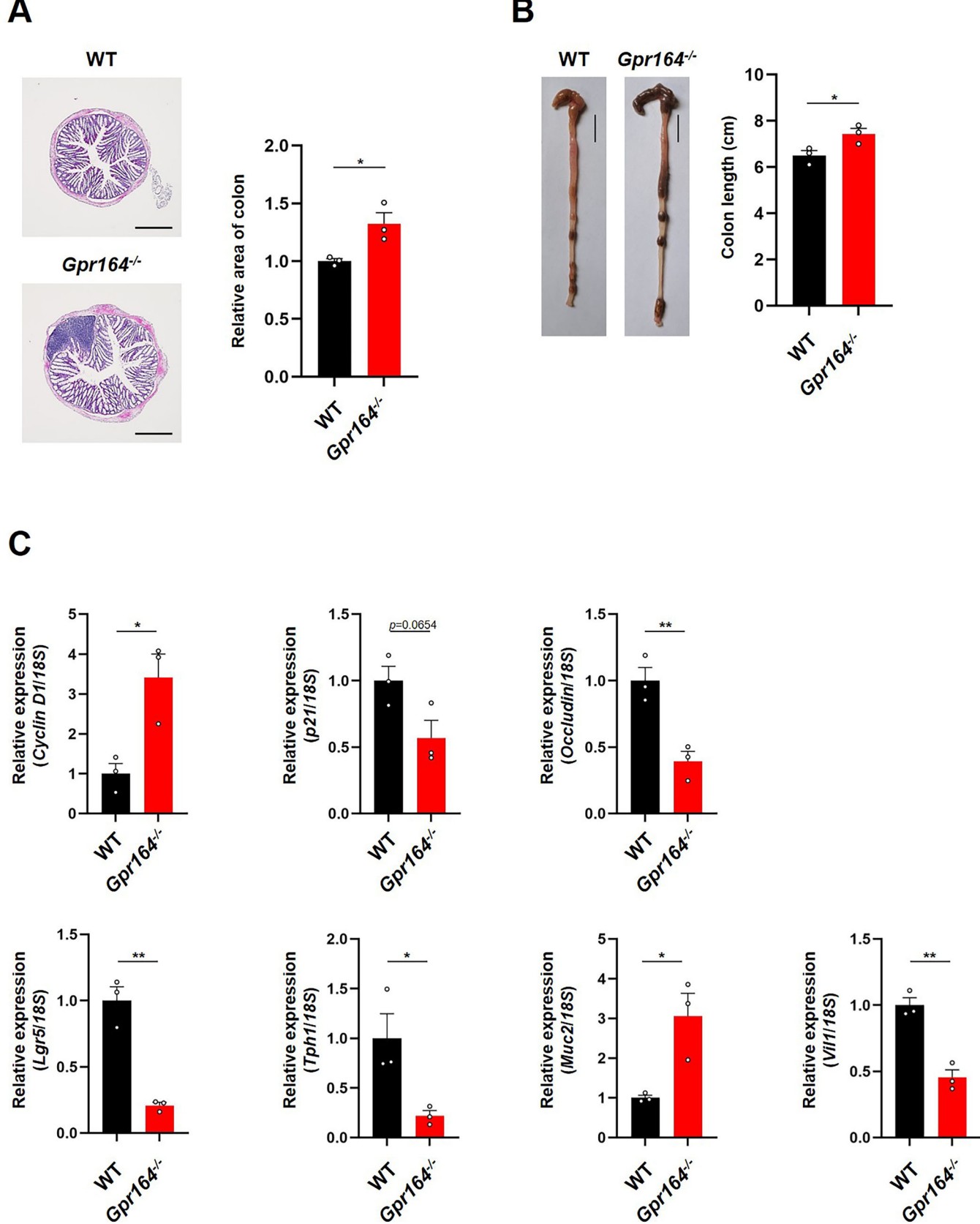

◀ **Figure EV2. Phenotypes of colonic hyperplasia in female *Gpr164*<sup>−/−</sup> mice.**

(A) Representative images of colon sections stained with hematoxylin-eosin (HE). The cross-sections of colon obtained from female WT and female *Gpr164*$^{-/-}$ mice were stained with HE, and the area of colon was measured using ImageJ software ($n = 3$). Scale bar, 500 μm. Error bars represent the mean ± SEM. *$P = 0.029$ (Student's *t*-test). (B) Representative images of colon ($n = 3$). Scale bar, 1 cm. Error bars represent the mean ± SEM. *$P = 0.0405$ (Student's *t*-test). (C) The expression levels of genes related to cell cycle, intestinal barrier and epithelial lineage were determined by qRT-PCR. Total RNA was extracted from colon tissue of female WT and female *Gpr164*$^{-/-}$ mice ($n = 3$). Error bars represent the mean ± SEM. *$P = 0.0194$; *Cyclin D1* gene, $P = 0.0654$; *p21* gene, **$P = 0.0083$; *Occludin* gene, **$P = 0.0018$; *Lgr5* gene, *$P = 0.0366$; *Tph1* gene, *$P = 0.023$; *Muc2* gene, **$P = 0.0024$; *Vil1* gene (*Tph-1* gene; Mann–Whitney *U*-test, other genes; Student's *t*-test). Source data are available online for this figure.

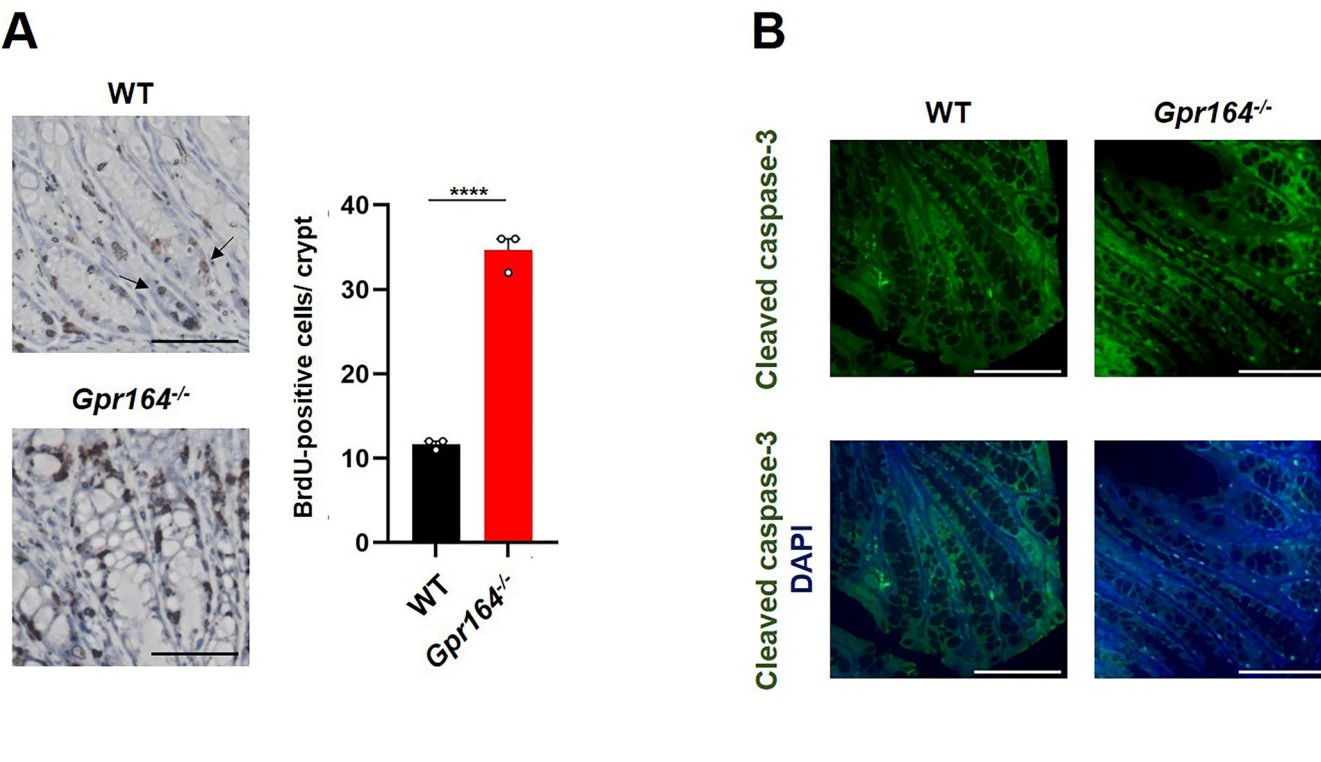

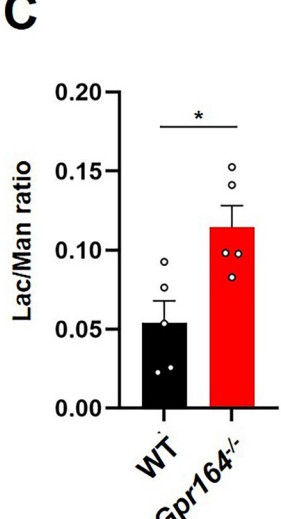

**Figure EV3. Effects of *Gpr164* deletion on proliferation, apoptosis, and intestinal barrier function.**

(A) Representative images of 5-bromo-2′-deoxyuridine (BrdU)-labeled cells in colon. The cross-sections of colon obtained from WT and Gpr164⁻/⁻ mice were stained with anti-BrdU antibody, and visualized with DAB (n = 3). Hematoxylin was used for nuclei staining. Scale bar, 50 μm. Error bars represent the mean ± SEM. ****$P < 0.0001$ (Student's $t$-test). (B) Representative images of immunofluorescent staining for cleaved caspase-3. The cross-sections of colon obtained from WT and Gpr164⁻/⁻ mice were stained with anti-cleaved caspase-3 antibody (n = 3). DAPI was used for nuclei staining. Scale bar, 50 μm. (C) Assessment of intestinal permeability by measuring urinary lactulose and mannitol. The excretion ratio of lactulose/mannitol in urine was assessed (n = 5). Error bars represent the mean ± SEM. *$P = 0.0143$ (Student's $t$-test). Source data are available online for this figure.

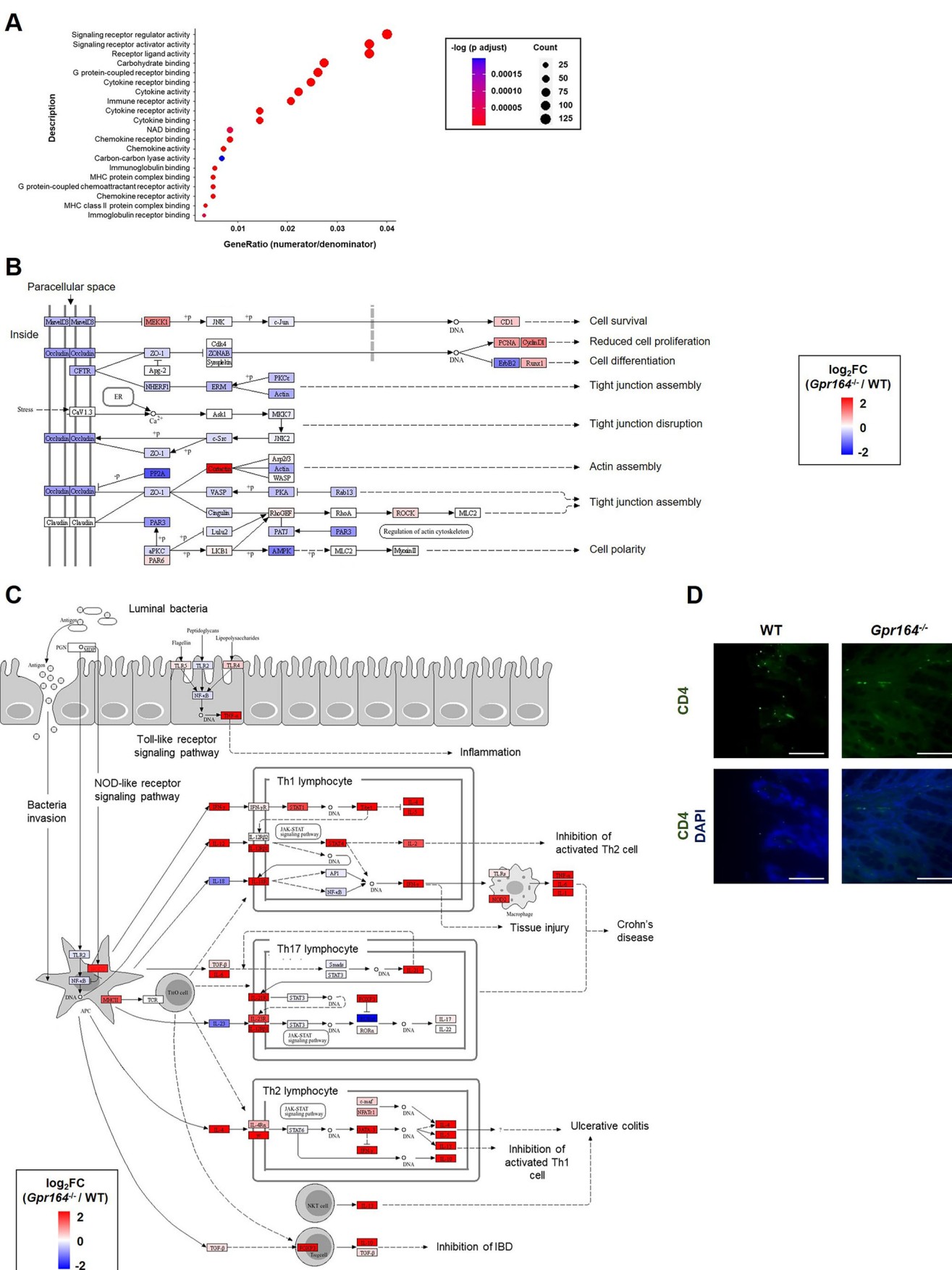

**Figure EV4.   Genome-wide RNA sequencing of *Gpr164*⁻ᐟ⁻ mice.**

(A) KEGG enrichment analysis involved in the molecular function in colon of *Gpr164*⁻ᐟ⁻ mice ($n = 5$). *P* values were adjusted based on the false discovery rate (FDR). (B, C) KEGG pathway enrichment related to tight junction (B) and inflammatory bowel disease (C). Increased or decreased levels of gene expressions are shown in red or blue, respectively. (D) Representative images of immunofluorescent staining for CD4. The cross-sections of colon obtained from WT and *Gpr164*⁻ᐟ⁻ mice were stained with anti-CD4 antibody ($n = 3$). DAPI was used for nuclei staining. Scale bar, 50 µm. Source data are available online for this figure.

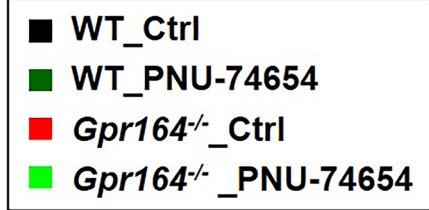

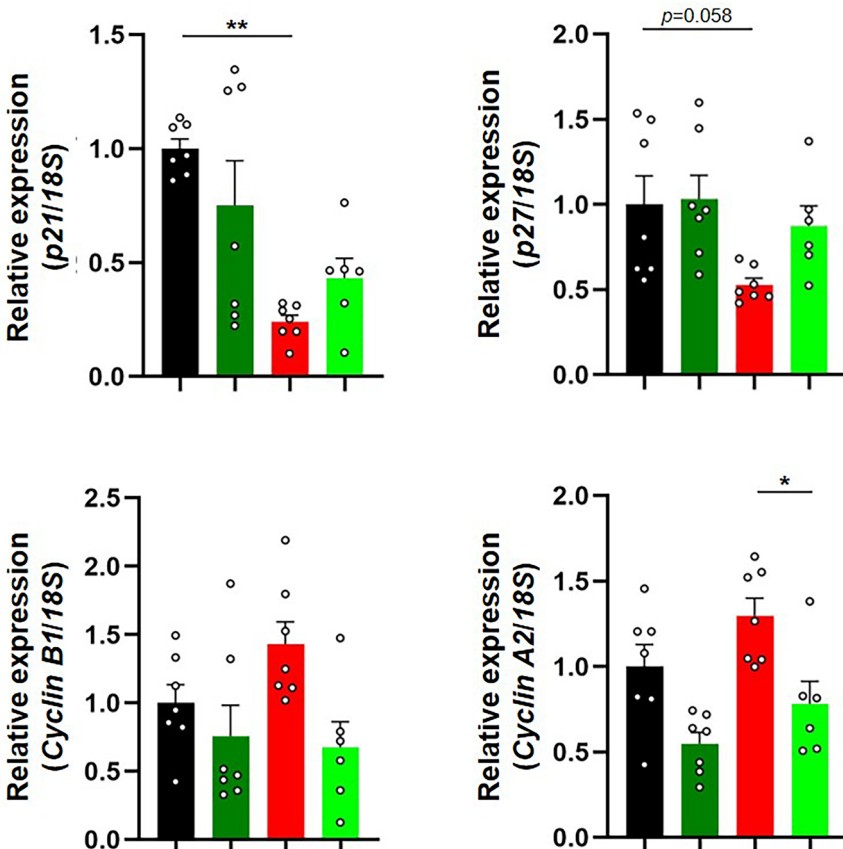

**Figure EV5.   Cell cycle gene expressions in PNU-74654-treated *Gpr164⁻ᐟ⁻* mice.**

The mRNA expression levels of cell cycle genes determined by qRT-PCR ($n = 6$–$7$). WT and $Gpr164^{-/-}$ mice were injected intraperitoneally with PNU-74654 (15 mg/kg body weight, every 2 days for 3 weeks), and total RNA was extracted from colon tissue of WT and $Gpr164^{-/-}$ mice ($n = 6$–$7$). Error bars represent the mean ± SEM. **$P = 0.0028$; WT_Ctrl vs $Gpr164^{-/-}$_Ctrl of $p21$ gene, $P = 0.0586$; WT_Ctrl vs $Gpr164^{-/-}$_Ctrl of $p27$ gene, *$P = 0.0173$; $Gpr164^{-/-}$_Ctrl vs $Gpr164^{-/-}$_ PNU-74654 of *Cyclin A2* gene ($p21$ and *Cyclin B1* genes; Dunn's test, $p27$ and *Cyclin A2* genes; Tukey–Kramer test). Source data are available online for this figure.

