## [Peer Review File · EMBO Reports]

The free fatty acid receptor GPR164 maintains intestinal homeostasis and barrier function

Takako Ikeda, Yuki Masujima, Keita Watanabe, Akari Nishida, Mayu Yamano, Miki Igarashi, Nobuo Sasaki, Hironori Katoh, and Ikuo Kimura

Corresponding author(s): Takako Ikeda (ikeda.takako.2r@kyoto-u.ac.jp), Ikuo Kimura (kimura.ikuo.7x@kyoto-u.ac.jp)

Review Timeline:

Submission Date:	24th Mar 25
Editorial Decision:	12th May 25
Revision Received:	4th Aug 25
Editorial Decision:	18th Sep 25
Revision Received:	25th Sep 25
Accepted:	5th Oct 25

Editor: Achim Breiling / Martina Rembold

Transaction Report:

Dear Dr. Ikeda,

Thank you for the submission of your manuscript to EMBO reports. I have now received the reports from the three referees that were asked to evaluate your study, which can be found at the end of this email.

As you will see, the referees think that these findings are of interest. However, they have several comments, concerns, and suggestions, indicating that a major revision of the manuscript is necessary to allow publication of the study in EMBO reports. As the reports are below, and all the referee concerns need to be addressed, I will not detail them here.

Given the constructive referee comments, I would like to invite you to revise your manuscript with the understanding that the concerns of the referees must be addressed in the revised manuscript and/or in a detailed point-by-point response. Acceptance of your manuscript will depend on a positive outcome of a second round of review. It is EMBO reports policy to allow a single round of revision only and acceptance of the manuscript will therefore depend on the completeness of your responses included in the next, final version of the manuscript.

1) a .docx formatted version of the final manuscript text (including legends for main figures, EV figures and tables), but without the figures included. Figure legends should be compiled at the end of the manuscript text.

2) individual production quality figure files as .eps, .tif, .jpg (one file per figure), of main figures and EV figures. Please upload these as separate, individual files upon re-submission.

4) a complete author checklist, which you can download from our author guidelines (<https://www.embopress.org/page/journal/14693178/authorguide>). Please insert page numbers in the checklist to indicate where the requested information can be found in the manuscript. The completed author checklist will also be part of the RPF.

5) that primary datasets produced in this study (e.g. RNA-seq, ChIP-seq, structural and array data) are deposited in an

appropriate public database. If no primary datasets have been deposited, please also state this in a dedicated section (e.g. 'No primary datasets have been generated and deposited'), see below.

The accession numbers and database should be listed in a formal "Data Availability" section that follows the model below. This is now mandatory (like the COI statement). Please note that the Data Availability Section is restricted to new primary data that are part of this study. This section is mandatory. As indicated above, if no primary datasets have been deposited, please state this in this section

Data availability

6) We now request the publication of original source data with the aim of making primary data more accessible and transparent to the reader. You will receive a separate email with instructions for providing source data with your revised manuscript, including information how to upload and organize the files.

8) Regarding data quantification and statistics, please make sure that the number "n" for how many independent experiments were performed, their nature (biological versus technical replicates), the bars and error bars (e.g. SEM, SD) and the test used to calculate p-values is indicated in the respective figure legends (also for EV and Appendix figures). Please also check that all the p-values are explained in the legend, and that these fit to those shown in the figure. Please provide statistical testing where applicable. Please avoid the phrase 'independent experiment', but clearly state if these were biological or technical replicates. Please also indicate (e.g. with n.s.) if testing was performed, but the differences are not significant. In case n=2, please show the data as separate datapoints without error bars and statistics. See also: <http://www.embopress.org/page/journal/14693178/authorguide#statisticalanalysis>

9) Please add scale bars of similar style and thickness to microscopic images, using clearly visible black or white bars (depending on the background). Please place these in the lower right corner of the images themselves. Please do not write on or near the bars in the image but define the size in the respective figure legend.

10) Please also note our reference format:

12) We now use CRediT to specify the contributions of each author in the journal submission system. CRediT replaces the author contribution section. Please use the free text box to provide more detailed descriptions and do NOT provide your final manuscript text file with an author contributions section. See also our guide to authors: <https://www.embopress.org/page/journal/14693178/authorguide#authorshipguidelines>

13) All Materials and Methods need to be described in the main text using our 'Structured Methods' format, which is required for

all research articles. According to this format, the Methods section should include a Reagents and Tools Table (listing key reagents, experimental models, software, and relevant equipment and including their sources and relevant identifiers), uploaded as separate file, and a Methods section in which we encourage the authors to describe their methods using a step-by-step protocol format with bullet points, to facilitate the adoption of the methodologies across labs. More information on how to adhere to this format as well as downloadable templates (.doc) for the Reagents and Tools Table can be found in our author guidelines (section 'Structured Methods'):

14) Please add up to five keywords to the manuscript and order the sections like this, using these names: Title page - Abstract - Keywords - Introduction - Results - Discussion - Methods - Data availability section - Acknowledgements - Disclosure and Competing Interests Statement - References - Figure legends - Expanded View Figure legends

15) Please make sure that all the funding information is also entered into the online submission system and that it is complete and similar to the one in the acknowledgement section of the manuscript text file.

I look forward to seeing a revised form of your manuscript when it is ready.

Yours sincerely,

Referee #1:

In their manuscript "The pivotal role of a novel free fatty acid receptor GPR164 in the intestinal barrier function" Ikeada and co-workers are describing a role for the novel free fatty acid receptor GPR164 in regulating colonic function. The authors describe a role for GPR164 in regulating intestinal cell proliferation and differentiation towards goblet cells, and conclude that loss of GPR164 results in severe inflammation. The manuscript is well written and contains interesting data with respect to the role of GPR164 in regulation of epithelial cell homeostasis. However, I do not agree with the authors interpretation that loss of GPR164 results in severe inflammation.

From what I can see in the images, the colonic epithelium looks intact, the muscle layer appears to be similar in thickness between genotypes, and there is no apparent increase in lamina propria immune cells. The observed increase in plasma TNFa and FITC dextran permeability is not in my opinion signs of severe inflammation. The observed increase in FITC dextran could be related to the increased frequency of goblet cells.

Although an increase in colonic crypt length is often a sign of increased bacterial contact with the epithelium, the authors show that the epithelial phenotype can be reversed by blocking wnt/b-catenin signaling, and it is in my opinion more likely that loss of GPR164 has a direct effect on epithelial proliferation rather than being a consequence of increased epithelial - bacterial interactions.

I think the authors have a nice set of data demonstrating a role for GPR164 in regulation of epithelial proliferation and differentiation. The manuscript would benefit from being restructured to focus on epithelial renewal and cancer rather than inflammation.

Below you can find more detailed comments to the manuscript.

Abstract: Page 2: I do not agree with the interpretation that these mice show signs of severe inflammation. Therefore, I do not

agree with the conclusion that this receptor is an attractive target for IBD treatment.

Introduction: Page 4 line 7: The authors should include goblet cells in the secretory cells, not only focusing on enteroendocrine cells. The authors discuss Butyrate's role in inflammation. However, Butyrate is a well-known HDAC inhibitor and regulator of epithelial proliferation. This should be discussed in the context of epithelial proliferation.

Results: Page 6 line 14: Supplementary figure 2 does not contain any images.

Page 6 line 22: The images in Fig. 2f are too small. It is not possible to evaluate the Ki67 staining from these pictures. The authors should quantify the Ki67 staining using images from multiple individuals. I also suggest that the authors add additional experiments to assess the number of proliferating cells using EdU injections.

Page 6 line 24-28: In this section the authors draw overreaching conclusions from their data. They have not showed any proof for a causative relationship between loss of mucus barrier integrity and an altered bacterial composition, and neither for the link between altered bacterial composition and elevated plasma TNF α levels. Please rephrase this statement. Furthermore, the authors have to quantify the mucus barrier defect in multiple individuals not just show one image per genotype. To make the claim that the mucus barrier is affected the authors have to do more than showing one image.

Page 7: The authors claim that there is a decreased abundance of Bacteroidota and increased abundance of Firmicutes in Gpr164 $^{-/-}$ mice but do not provide statistical support for this claim. Fig. 3e shows the average abundance of Bacteroidata and Firmicutes. Please add statistical support for the statement.

Page 8: The colon does not have villi, please rephrase. The authors suggest that the reduced expression of Lgr5 could be due to stem cells not receiving enough Wnt to maintain proliferation. Since the authors show a massive increase in crypt length, an alternative explanation may be that the proportion of Lgr5 $^{+}$ stem cells is reduced in relation to the total epithelial cell number. The authors should assess the number of Lgr5 $^{+}$ stem cells in WT and GPR164 $^{-/-}$ mice.

Page 8: Fig. 4c and d). I assume that the ectopic expression of B-catenin that the authors refer to is nuclear localization of B-catenin in the surface epithelium of GPR164 $^{-/-}$ mice. It is not possible to determine the localization of the staining based on these images. This analysis has to be redone at a higher resolution to allow the reader to determine the localization of the signal.

Page 8: Supplementary figure 3a,b, c). Sup. 3a is not referred to in the text. Please refer to this figure in the text or remove it. The resolution of supp fig. 3c is low and it is difficult to read some of the gene names. These flow charts are barely referred to in the text and they need some more context. Please add more information to these flow charts to put the results in the context of the paper. Supp Fig. 3c) suggests that there is an increase in effector T cells in the colon LP of GPR164 $^{-/-}$ mice. To support this finding, it would be good if the authors could provide additional data such as flow cytometry on isolated colon lamina propria cells or immunostainings of colonic tissues.

Page 9, line 13-15: It is not possible to evaluate the Ki67 staining in these small images. Please redo the staining and provide images at a higher resolution. The data should also be quantified to determine the effect of the treatment on epithelial cell proliferation.

Page 10 line 9 to 18: The FITC dextran experiments on the organoids have to be quantified. Showing one image per treatment is not enough to draw any conclusions on the effect of the different treatments.

Methods: Page 16: Why were only male mice used in this study? Were WT and GPR164 $^{-/-}$ mice co-housed during the different experiments?

Page 17: HE staining and immunohistochemistry. Please add the type of antigen retrieval used.

Discussion: Please tone down the causative relationship between your different findings. Although you have made several observations, this does not mean that they are all directly linked.

Statistics:

Figure 1d and 1f): When measuring data over time a two-way ANOVA is more appropriate than repeated t-test.

General comment: I think it would benefit the manuscript to include data demonstrating which epithelial cell type(s) that express GPR164. This can be done using either RNAscope or immunostainings.

Referee #2:

I have reviewed this manuscript and believe the results shown are a reliable representation of the experiments performed by Ikeda et al.

I must say that the English in the opening paragraph is far from easy to read and I would encourage the authors to improve the English. However, the introduction wanders through too many topics and should be focussed on the main findings. Where the authors state a fact e.g. "Although a large number of FFAs such as iso-butyrate, valerate and nonanoate has (have ?) been identified as ligands for GPR16" there should be a reference to the primary data.

Some comment on the viability and breeding of the mice should be given and evidence that gpr164 protein expression is absent would be helpful.

Although the authors claim to measure changes in proliferation, their measurements are surrogates for this process, at the least they should acknowledge that changes in cell death processes could contribute to the increased cellularity of the crypts and area of the colonic mucosa.

Whilst many of the changes in parameters are marginal, the evidence for a change in the intestinal barrier function is strong. It might have been helpful to have a lactulose or mannitol barrier test, as the increased levels of TNF-alpha are more likely to be a measure of increased inflammation. Which could be caused by infections etc.

The results of the organoid permeability test with the FITC-Dextran look convincing, although quantitation and the reliability of the results should be presented/ The roles of butyrate and palmoate in that experiment should be explained.

The results section contains many conclusions and observations which are not results, but discussion. It would be a clearer presentation if the results were presented and then the discussion section presented the interpretation and ramifications of the results

Referee #3:

The paper addresses the function of the fatty acid binding G-protein coupled receptor Gpr164 in the intestine of WT and Gpr164-KO mice. The topic is important as nutrient and microbially produced fatty acids can affect intestinal physiology/immunology - and Gpr164 (as shown here) is expressed throughout the mouse gut.

The report shows good evidence for Wnt-dependent intestinal proliferation and enhanced goblet cell expression in Gpr164-KO mice. This is associated with decreased expression of several intracellular tight junction proteins and enhanced passive solute flux indicative of altered barrier function - i.e. more permeable to passive diffusion of luminal solutes. The barrier defect induced by Gpr164-KO was also observed in a human intestinal cell line and in mouse intestinal organoids. Alteration of the gut microbiome is measured in KO mice - and this is attributed to loss of the extra cellular mucus layer in the colon. But this data is weak (the imaging is not clear and mucus layer thickness should be quantified). The claim is also counterintuitive as the goblet cell populations are increased. There is good evidence for increased TNFa secretion associated with loss of Gpr164. This comes with the claim that the TNFa response results from the defective mucus layer, loss of barrier function, and dysbiosis - but this was not tested. Convincing data using a WNT inhibitor, however, shows the TNFa phenotype likely depends on high WNT induced by deletion of Gpr164-KO - and this is linked to the goblet cell/microbiome phenotypes.

Overall, the claim that Gpr164 operates in the intestine to affect WNT-dependent phenotypes is reasonably supported. Suggested mechanisms of action however remain inconclusively tested.

Additional Comments:

Figure 1: Wildtype Caco cells are not the best control here-Caco cells transfected with Cas9 alone or Cas9 with an off-target sgRNA would be preferable.

Figure 3: Several microbiome changes have been associated with impaired mucus and epithelial barrier function (eg. Akkermansia and Bifidobacteria depletion, loss of several butyrate producers). This raises the possibility that some of the phenotype might be microbiome-driven. This is worth discussing.

Figure 6: FITC-dextran study in organoids is nice. It complements the Caco TER data in Figure 1 and demonstrates that the butyrate effect on barrier integrity is GPR164-dependent.

It would be helpful to have a quantitative measure, or at least present data from more than one organoid per condition, to increase the robustness of this finding.

Butyrate increases TJ protein expression and TER in Caco cells as well, but this was not done. If GPR164 KO Caco cells do not similarly respond to butyrate, this would significantly increase the rigor of the author's conclusion that GPR164 is necessary for butyrate-mediated epithelial barrier function is GPR164 dependent.

The Senior Editor
EMBO Reports

Dear Dr. Breiling

Thank you very much for your e-mail (May 12, 2025) concerning our manuscript (EMBOR-2025-61597V1) entitled "The pivotal role of a novel free fatty acid receptor GPR164 in the intestinal barrier function". We revised our manuscript to meet the reviewers' comments, and herewith enclosed the revised manuscript. We addressed all the points raised by three reviewers, as described below (Point-by point response). Also, we modified the revised manuscript in accordance with the journal's format requirements.

We updated author affiliation of Dr. Igarashi in the revised manuscript.

We hope that this manuscript is now acceptable for publication in *EMBO Reports*. We are most grateful to you and reviewers for helpful comments.

Sincerely yours,

Takako Ikeda
Ikuo Kimura

Laboratory of Molecular Neurobiology, Graduate School of Biostudies, Kyoto University.

Sakyo-ku, Kyoto 606-8501, Japan.

Tel; (+81) 75-753-4547

email: Takako Ikeda (ikeda.takako.2r@kyoto-u.ac.jp)

Ikuo Kimura (kimura.ikuo.7x@kyoto-u.ac.jp)

Response to the reviewers' comments

(Point-by point response)

Referee #1:

Comments: In their manuscript "The pivotal role of a novel free fatty acid receptor GPR164 in the intestinal barrier function" Ikeada and co-workers are describing a role for the novel free fatty acid receptor GPR164 in regulating colonic function. The authors describe a role for GPR164 in regulating intestinal cell proliferation and differentiation towards goblet cells, and conclude that loss of GPR164 results in severe inflammation. The manuscript is well written and contains interesting data with respect to the role of GPR164 in regulation of epithelial cell homeostasis. However, I do not agree with the authors interpretation that loss of GPR164 results in severe inflammation.

From what I can see in the images, the colonic epithelium looks intact, the muscle layer appears to be similar in thickness between genotypes, and there is no apparent increase in lamina propria immune cells. The observed increase in plasma TNF α and FITC dextran permeability is not in my opinion signs of severe inflammation. The observed increase in FITC dextran could be related to the increased frequency of goblet cells.

Responses: According to the referee's comments, we performed immunofluorescence analysis using an anti-CD4 antibody and found the increased CD4 positive cells in lamina propria (new Figure EV4). In addition to this, some mice exhibit an increased thickness of the muscle layer (shown in below). However, as the referee #1 mentioned, the phenotypes observed in *Gpr164*^{-/-} mice do not appear to be severe inflammation. Therefore, we have changed the word "severe inflammation" to "inflammation" in our revised manuscript.

Comments: Although an increase in colonic crypt length is often a sign of increased bacterial contact with the epithelium, the authors show that the epithelial phenotype can be reversed by blocking wnt/b-catenin signaling, and it is in my opinion more likely that loss of GPR164 has a direct effect on epithelial proliferation rather than being a consequence of increased epithelial - bacterial interactions.

I think the authors have a nice set of data demonstrating a role for GPR164 in regulation of epithelial proliferation and differentiation. The manuscript would benefit from being restructured to focus on epithelial renewal and cancer rather than inflammation.

Responses: According to the referee's suggestions, we restructured our manuscript to focus on epithelial renewal and cancer. We appreciate your insightful comments.

Comments: Below you can find more detailed comments to the manuscript.

Abstract: Page 2: I do not agree with the interpretation that these mice show signs of severe inflammation. Therefore, I do not agree with the conclusion that this receptor is an attractive target for IBD treatment.

Responses: According to the referee's suggestions, we rewrote the sentences in the abstract section of our revised manuscript as described below.

"*Gpr164*^{-/-} mice also exhibited gut microbial dysbiosis and inflammation. Thus, our findings uncover the pivotal role of GPR164 in the maintenance of intestinal homeostasis through regulating the barrier function." (new lines; 47-50).

Comments: Introduction: Page 4 line 7: The authors should include goblet cells in the secretory cells, not only focusing on enteroendocrine cells. The authors discuss Butyrate's role in inflammation. However, Butyrate is a well-known HDAC inhibitor and regulator of epithelial proliferation. This should be discussed in the context of epithelial proliferation.

Responses: According to the referee's suggestions, we included goblet cells in the

secretory cells (new line 110). Also, we described that butyrate acts as HDAC inhibitor and regulates the intestinal epithelial growth in the introduction section of our revised manuscript as described below.

“Butyrate is also involved in the regulation of intestinal epithelial cell growth through inhibiting histone deacetylase (HDAC) activity¹². Although butyrate is rapidly used as fuel in the colon, intake of non-digestive carbohydrates or the progression of colorectal cancer leads to an increased accumulation of butyrate, which allows butyrate to act as a signaling molecule for FFARs and an HDAC inhibitor. Inhibition of HDAC activity by butyrate resulted in hyperacetylation of histone H3, leading to the transcriptional activation of genes associated with cell cycle inhibition and apoptosis¹². Therefore, anti-proliferative property of butyrate through HDAC inhibition has gained attention for the treatment of colorectal cancer (new lines 117-125).”

Comments: Results: Page 6 line 14: Supplementary figure 2 does not contain any images.

Responses: In the revised manuscript, we corrected figure number which the referee#1 mentioned. Thank you for the referee’s kind comment.

Comments: Page 6 line 22: The images in Fig. 2f are too small. It is not possible to evaluate the Ki67 staining from these pictures. The authors should quantify the Ki67 staining using images from multiple individuals. I also suggest that the authors add additional experiments to assess the number of proliferating cells using EdU injections.

Responses: In accordance with the referee’s suggestions, we enlarged the images of Ki67 immunostaining and quantified the Ki67-positive epithelial cells (new Fig. 2F). We also performed BrdU incorporation assay and found increased BrdU incorporation in colonic epithelial cells of *Gpr164*^{-/-} mice (new Fig. EV3A). We think that these data strengthen our findings. We greatly appreciate the referee’s helpful suggestion.

Comments: Page 6 line 24-28: In this section the authors draw overreaching conclusions from their data. They have not showed any proof for a causative relationship between loss of mucus barrier integrity and an altered bacterial composition, and neither for the link between altered bacterial composition and elevated plasma TNFa levels. Please rephrase this statement. Furthermore, the authors have to quantify

the mucus barrier defect in multiple individuals not just show one image per genotype. To make the claim that the mucus barrier is affected the authors have to do more than showing one image.

Responses: In accordance with the referee's suggestions, we rephrased the sentences as described below.

“Taken together, these results indicate that GPR164 is involved in the regulation of the mucus barrier, microbial composition, SCFA production and inflammation. (new lines; 245-247).”

In addition, according to the referee's suggestions, we quantified the thickness of mucin layer in multiple individuals (new Fig. 3B).

Comments: Page 7: The authors claim that there is a decreased abundance of Bacteroidota and increased abundance of Firmicutes in *Gpr164*^{-/-} mice but do not provide statistical support for this claim. Fig. 3e shows the average abundance of Bacteroidata and Firmicutes. Please add statistical support for the statement.

Responses: According to the referee's suggestions, we performed qRT-PCR to quantify the abundance of Firmicutes and Bacteroides in the gut microbiota (new Fig. 3E). The abundance of Firmicutes in *Gpr164*^{-/-} mice was increased with statistical significance compared to WT mice under normal diet conditions.

Comments: Page 8: The colon does not have villi, please rephrase. The authors suggest that the reduced expression of *Lgr5* could be due to stem cells not receiving enough Wnt to maintain proliferation. Since the authors show a massive increase in crypt length, an alternative explanation may be that the proportion of *Lgr5*⁺ stem cells is reduced in relation to the total epithelial cell number. The authors should assess the number of *Lgr5*⁺ stem cells in WT and *GPR164*^{-/-} mice.

Responses: As the referee#1 pointed out, the colon does not have villi. We rephrased it in the revised manuscript (new lines 275-281). Also, according to the referee's suggestions, we performed immunohistochemistry for *Lgr5* on colon sections. However, we unexpectedly found strong staining along the crypt axis (new Fig. EV4B). It indicates that there seems to be non-specific staining of *Lgr5* because *Lgr5* is generally

localized in the crypt base. Therefore, I regret to inform you that our Lgr5 immunostaining result cannot provide insight into the question about the effect of GPR164 on the proportion of Lgr5 positive stem cells in relation to the total epithelial cell number. I appreciate your positive and thoughtful comments.

Comments: Page 8: Fig. 4c and d). I assume that the ectopic expression of B-catenin that the authors refer to is nuclear localization of B-catenin in the surface epithelium of GPR164^{-/-} mice. It is not possible to determine the localization of the staining based on these images. This analysis has to be redone at a higher resolution to allow the reader to determine the localization of the signal.

Responses: According to the referee's suggestions, we showed the images with higher resolution in the revised figures (new Figs. 4C, D). As the referee suggested, nuclear localization of β -catenin in the epithelial cells of GPR164^{-/-} mice was observed. We added the sentences as described below. We appreciate referee's important suggestion.

“A heatmap of the gene expression profile revealed that the expression levels of many genes associated with Wnt signaling pathway were changed markedly, and aberrant expression of β -catenin in the top of crypt was observed in *Gpr164*^{-/-} mice (Figs. 4C, D). Besides, nuclear localization of β -catenin in the epithelial cells of *Gpr164*^{-/-} mice was also observed (Fig. 4D, right). Thus, these findings suggest that *Gpr164* deficiency results in both aberrant expression and ectopic localization of β -catenin in the epithelial cells, which causes abnormal proliferation (colonic hyperplasia) and differentiation (increased goblet cells and reduced enteroendocrine/absorptive cells) of colonic epithelial cells. (new lines 278-286).”

Comments: Page 8: Supplementary figure 3a,b, c). Sup. 3a is not referred to in the text. Please refer to this figure in the text or remove it. The resolution of supp fig. 3c is low and it is difficult to read some of the gene names. These flow charts are barely referred to in the text and they need some more context. Please add more information to these flow charts to put the results in the context of the paper. Supp Fig. 3c) suggests that there is an increase in effector T cells in the colon LP of GPR164^{-/-} mice. To support this finding, it would be good if the authors could provide additional data such as flow cytometry on isolated colon lamina propria cells or immunostainings of colonic tissues.

Responses: According to the referee's suggestions, Sup. 3a (new Fig. EV4A) is referred

to in the text as described below.

“KEGG enrichment analysis revealed that the expression of genes related to the regulation of receptor activities were significantly changed in *Gpr164*^{-/-} mice (Fig. EV4A) (new lines 262-264).”

Also, we showed Sup. 3c with higher resolution in the revised figures (new Fig. EV4D). To evaluate the effect of *Gpr164* deletion on effector T cells, we performed immunofluorescent staining for CD4. In the colon lamina propria of *Gpr164*^{-/-} mice, CD4 positive T cells were found (new Fig. EV4E). In accordance with the referee’s suggestions, we added this data and referred to the results of new Figs. EV4C and E in the revised manuscript as described below.

“Consistent with the data obtained in Fig. 2H and Fig. 3F, Kyoto Encyclopedia of Genes and Genome (KEGG) analysis showed that *Gpr164* deletion leads to disruption of epithelial barrier integrity and promotion of inflammation in the colon (Figs. EV 4C, D). In *Gpr164*^{-/-} mice, many genes involved in the tight junction assembly and cell polarity were reduced, whereas several marker genes for effector T cells were increased (Figs. EV 4C, D). To confirm findings of the data shown in Fig. EV 4D, we performed immunofluorescent staining for CD4 and found an increase in CD4 positive cells in the colon lamina propria of *Gpr164*^{-/-} mice (Fig. EV 4E) (new lines 294-301).”

Comments: Page 9, line 13-15: It is not possible to evaluate the Ki67 staining in these small images. Please redo the staining and provide images at a higher resolution. The data should also be quantified to determine the effect of the treatment on epithelial cell proliferation.

Responses: According to the referee’s suggestions, we showed the images with higher resolution and quantified the Ki67-positive epithelial cells (new Fig. 5D).

Comments: Page 10 line 9 to 18: The FITC dextran experiments on the organoids have to be quantified. Showing one image per treatment is not enough to draw any conclusions on the effect of the different treatments.

Responses: According to the referee’s suggestions, we quantified the fluorescence intensities of FITC-dextran (new Fig. 6E).

Comments: Methods: Page 16: Why were only male mice used in this study? Were WT and GPR164^{-/-} mice co-housed during the different experiments?

Responses: WT and *Gpr164*^{-/-} mice were not co-housed. In accordance with the referee's suggestion, we examined the morphological changes in the colon of female *Gpr164*^{-/-} mice. HE staining revealed an increase in both area and length of colon (new Fig. EV2A, B). Also, the mRNA expression levels of genes associated with cell cycle progression, tight junction and epithelial lineages in female *Gpr164*^{-/-} mice were similar to those in male *Gpr164*^{-/-} mice (new Fig. EV2C). Thus, GPR164 plays an important role in the maintenance of intestinal homeostasis in a sex-independent manner. We added the data and sentences in the revised manuscript as described below.

“*Gpr164*^{-/-} female mice also exhibited the hyperplasia phenotype in the colon such as an increased volume and cell cycle progression (Figs. EV 2A, B, C). Furthermore, decreased expression of tight junction marker genes was observed in both male and female *Gpr164*^{-/-} mice (Fig. 2G and Fig. EV2C) (new lines 198-201).”

“In addition, quantitative PCR analysis showed that the expression of *Tph-1* and *Vill* was decreased but that of *Muc2* was increased in both male and female *Gpr164*^{-/-} mice, confirming that a loss of *Gpr164* resulted in the enhanced differentiation into goblet cells (Fig. 4B and Fig. EV2C) (new lines 266-270).”

Comments: Page 17: HE staining and immunohistochemistry. Please add the type of antigen retrieval used.

Responses: In accordance with the referee's comment, we added the type of antigen retrieval used in this study in the material and methods sections as described below.

“The cross-section was heated in 10 mM sodium citrate buffer (pH 6.0) for 50 min to retrieve antigen (new line 533-534).”

Comments: Statistics: Discussion: Please tone down the causative relationship between your different findings. Although you have made several observations, this does not mean that they are all directly linked.

Responses: As the referee mentioned, our findings did not show a direct link between

phenotype. We therefore rewrote the manuscript to tone down the causative relationship between our different findings. We appreciate your considerable comments.

Comments: Statistics: Figure 1d and 1f): When measuring data over time a two-way ANOVA is more appropriate than repeated t-test.

Responses: In accordance with the referee's suggestion, we analyzed the data shown in new Fig. 1D and 1F by using a two-way ANOVA with sidak's post hoc test.

Comments: General comment: I think it would benefit the manuscript to include data demonstrating which epithelial cell type(s) that express GPR164. This can be done using either RNAscope or immunostainings.

Responses: In accordance with the referee's suggestion, we performed immunofluorescent staining using commercially available antibody for GPR164. The fluorescent signals of GPR164 were detected in the colon sections of WT mice but not of *Gpr164*^{-/-} mice (Fig. EV1G). Consistent with a previous report (*), signals were mostly detected at the crypt base. Also, we reanalyzed the dataset (GSE229814) uploaded in 2023 (*) by using the Seurat package in R studio. After data normalization, dimension reduction and clustering, we found that GPR164 was enriched in cluster 14 as shown below. Using the PanglaoDB database (<https://panglaodb.se/index.html>), we determined which cell types express the genes classified as cluster 14. The genes in cluster 14 (shown below), except for *Fhl5*, mostly express in pericytes or smooth muscle cells. Therefore, these data indicate the possibility that GPR164 expresses in pericytes or smooth muscle cells.

(*) Dinsart G, Leprovots M, Lefort A, Libert F, Quesnel Y, Veithen A, Vassart G, Huyseune S, Parmentier M, Garcia MI (2023) The olfactory receptor Olfr78 promotes differentiation of enterochromaffin cells in the mouse colon. *EMBO Rep.* 25: 304-333

cluster	genes									
0	Tph1	Gpr37l1	S100b	Chga	Kcna1	Sfrp5	Ptprz1	Cdh19	Gfap	Resp18
1	Edil3	Hapln1	Col6a5	Dnmt3l	Cldn1	Aldh3a1	Frzb	Ano5	Prtn3	Frmpd4
2	Slc51a	Abca12	Slc10a2	Arg2	Acaa1b	Cyp3a44	Ggt1	Cyp3a25	Akr1c18	Reg3a
3	Krt6a	Trp63	Dapl1	Krtdap	Krt13	Krt1	Calm4	Gm94	Spink5	Lgals7
4	Ly6g	Tgm3	Anxa8	10079G19l	Fxyd4	Csta1	Gjb4	10034C09f	Syt8	Trpv6
5	Glp2r	Fgf9	Cybrd1	Trpa1	Ano2	Fst	Rhox5	Adra2c	Dmp1	Spp1
6	Cilp	Pcolce2	Ackr4	Inmt	Pi16	Gdf10	Sult1e1	Myoc	Itgbl1	Kera
7	Best2	Scgb1b3	Cma1	Ang4	Frmd3	30020B18f	Ces1c	Tmprss13	Serpina9	Mmp7
8	Atp12a	Slc37a2	Ccdc152	Saa2	H60c	Mettl7a2	Gal3st2b	Fmo4	Susd1	Ceacam2
9	Flt1	Fabp4	Plvap	Thrsp	Sox17	Exoc3l2	C1qtnf9	Aplnr	Esm1	Selp
10	Csf1r	Cx3cr1	Cd300ld	Ccl3	Fcgr4	Aoah	Fcgr1	Mmp13	Lyz1	Il22ra2
11	Mmrn1	Ccl21a	Lyve1	Reln	Lcn2	Gpm6a	Adgrg3	Tbx1	Pkhd11l	Htr2b
12	Actg2	Cnn1	Hhip	Cdh6	Frem2	Sntg2	Bves	Actc1	Chrdl2	Ckm
13	Cd3e	Xcr1	Cd8a	Flt3	Siglecg	Ms4a4b	Cxcr6	Ccl22	Gzmb	Cd160
14	Notch3	Ndufa4l2	Vtn	Rasl12	Map3k7cl	Nrip2	Pln	Colec11	Scn4b	Fhl5
15	Derl3	Tnfrsf17	Ccr10	Eaf2	Cacna1s	Ly6c2	Prg2	Sipi	Oosp1	Jchain

Referee #2:

Ms EMBOR-2025-61597V1

Comments: I have reviewed this manuscript and believe the results shown are a reliable

representation of the experiments performed by Ikeda et al. I must say that the English in the opening paragraph is far from easy to read and I would encourage the authors to improve the English. However, the introduction wanders through too many topics and should be focussed on the main findings.

Responses: In accordance with the referee's suggestion, we rewrote the opening paragraph to improve our manuscript as described below. Also, we reduced topics in the introduction section of our revised manuscript to focus on the main findings.

"GPR164 was initially identified as an olfactory chemosensory receptor. GPR164 is also known as Olfr558 in mouse and OR51E1 in human, and Olfr558 and OR51E1 are encoded by *Or51e1* and *OR51E1* gene, respectively. GPR164 is expressed in several tissues including the olfactory epithelium. In peripheral tissues, GPR164 is activated by free fatty acids (FFAs) such as butyrate and nonanoate, thereby being considered as a free fatty acid receptor (FFAR)^{1,2}. (new lines 74-79)."

Comments: Where the authors state a fact e.g. "Although a large number of FFAs such as iso-butyrate, valerate and nonanoate has (have ?) been identified as ligands for GPR16" there should be a reference to the primary data.

Responses: As the referee suggested, we added the references.

Comments: Some comment on the viability and breeding of the mice should be given and evidence that *gpr164* protein expression is absent would be helpful.

Responses: In accordance with the referee's suggestion, we added the following sentence in the Methods section in our revised manuscript.

"*Gpr164*^{-/-} mice are viable and have no obvious abnormalities in behavior and fertility (new lines 456-457)."

For the detection of GPR164, we performed immunofluorescent analysis. As shown in new Fig. EV1G, fluorescent signals of GPR164 were not detected in the colon of *Gpr164*^{-/-} mice.

Comments: Although the authors claim to measure changes in proliferation, their

measurements are surrogates for this process, at the least they should acknowledge that changes in cell death processes could contribute to the increased cellularity of the crypts and area of the colonic mucosa.

Responses: In accordance with the referee's suggestion, we examined the effect of GPR164 on apoptosis by immunofluorescence staining using anti-cleaved caspase-3 antibody. As shown in new Fig. EV3B, no obvious differences between genotypes were observed. Also, there was no significant changes in p53 protein expression in control and *OR51E1* KO Caco-2 cells (new Fig. 1G). We greatly appreciate the referee's important suggestion.

Comments: Whilst many of the changes in parameters are marginal, the evidence for a change in the intestinal barrier function is strong. It might have been helpful to have a lactulose or mannitol barrier test, as the increased levels of TNF-alpha are more likely to be a measure of increased inflammation. Which could be caused by infections etc.

Responses: In accordance with the referee's suggestion, we performed lactulose/mannitol test. As shown in new Fig. EV3C, the lactulose/mannitol ratio was significantly increased in *Gpr164*^{-/-} mice, indicating that the integrity of the intestinal barrier function was impaired. This data strengthens our findings. We greatly appreciate the referee's helpful suggestion.

Comments: The results of the organoid permeability test with the FITC-Dextran look convincing, although quantitation and the reliability of the results should be presented/ The roles of butyrate and palmoate in that experiment should be explained.

Responses: In accordance with the referee's suggestion, we quantified the fluorescence intensities of FITC-dextran (new Fig. 6E). The role of butyrate and palmitate in the experiments using organoid was described as follows.

“Butyrate is a bioactive product generated through microbial fermentation, and strengthens both tight junction and mucus barrier functions. Given that butyrate is a dominant ligand for GPR164 in colon, butyrate-mediated enhancement of the intestinal barrier function may be exerted through GPR164 activation (new lines 330-333).”

“Palmitate has been reported to alter the expression of Hes1 and Muc2, thereby influencing the differentiation of certain types of cells such as goblet cells³⁰. Therefore,

besides pro-inflammatory property²⁹, palmitate induces lipotoxicity through impairment of epithelial differentiation (new lines 353-356).”

Comments: The results section contains many conclusions and observations which are not results, but discussion. It would be a clearer presentation if the results were presented and then the discussion section presented the interpretation and ramifications of the results

Responses: In accordance with the referee’s comments, we clearly described the results. Also, the interpretation and ramifications of the results were described in the discussion section.

Referee #3:

Comments: The paper addresses the function of the fatty acid binding G-protein coupled receptor Gpr164 in the intestine of WT and Gpr164-KO mice. The topic is important as nutrient and microbially produced fatty acids can affect intestinal

physiology/immunology - and Gpr164 (as shown here) is expressed throughout the mouse gut.

The report shows good evidence for Wnt-dependent intestinal proliferation and enhanced goblet cell expression in Gpr164-KO mice. This is associated with decreased expression of several intracellular tight junction proteins and enhanced passive solute flux indicative of altered barrier function - i.e. more permeable to passive diffusion of luminal solutes. The barrier defect induced by Gpr164-KO was also observed in a human intestinal cell line and in mouse intestinal organoids. Alteration of the gut microbiome is measured in KO mice - and this is attributed to loss of the extra cellular mucus layer in the colon. But this data is weak (the imaging is not clear and mucus layer thickness should be quantified). The claim is also counterintuitive as the goblet cell populations are increased. There is good evidence for increased TNF α secretion associated with loss of Gpr164. This comes with the claim that the TNF α response results from the defective mucus layer, loss of barrier function, and dysbiosis - but this was not tested. Convincing data using a WNT inhibitor, however, shows the TNF α phenotype likely depends on high WNT induced by deletion of Gpr164-KO - and this is linked to the goblet cell/microbiome phenotypes.

Overall, the claim that Gpr164 operates in the intestine to affect WNT-dependent phenotypes is reasonably supported. Suggested mechanisms of action however remain inconclusively tested.

Responses: In accordance with the referee's comments, we quantified the thickness of mucin layer in multiple individuals (new Fig. 3B). As the referee mentioned, our findings did not show a direct link between dysbiosis and inflammation. We therefore rewrote the manuscript to tone down the causative relationship between our different findings.

Comments: Additional Comments:

Figure 1: Wildtype Caco cells are not the best control here-Caco cells transfected with Cas9 alone or Cas9 with an off-target sgRNA would be preferable.

Responses: In accordance with the referee's suggestions, we generated control KO cells by transfection of Caco-2 cells with peSpCAS9(1.1)-2xsgRNA vector alone. Using control KO cells, we demonstrated quantitative RT-PCR analyses, crystal violet staining,

TER measurement, western blotting and PCR (new Figs. 1C-G, 6D-E and Figs. EV 1B-C).

Comments: Figure 3: Several microbiome changes have been associated with impaired mucus and epithelial barrier function (eg. Akkermansia and Bifidobacteria depletion, loss of several butyrate producers). This raises the possibility that some of the phenotype might be microbiome-driven. This is worth discussing.

Responses: In accordance with the referee's suggestions, we discussed the possibility that some of the phenotype might be driven by the changes in microbiome in the discussion section of our revised manuscript as described below.

“Our 16S rRNA sequencing revealed that the presence of *Akkermansiaceae*, *Bifidobacteriaceae*, *Prevotellaceae* in the colon of *Gpr164*^{-/-} mice tended to decrease (Fig. 3D). In particular, *Akkermansiaceae*, mainly *Akkermansia muciniphila* (*A. muciniphila*), is known to be associated with inflammatory diseases and cancer. *A. muciniphila* degrades mucin in the colon, which causes inflammation and the subsequent development of colorectal cancer²⁵. Contrary to this, a marked increase in the abundance of *A. muciniphila* is observed in the patients treated with chemotherapy for colorectal cancer, and its abundance is positively correlated with the therapeutic effect²⁵. These reports suggest that the abundance of *A. muciniphila* is strongly associated with colorectal cancer, and hence GPR164 might be involved in the development of colorectal cancer through affecting gut microbial composition. Additionally, changes in the luminal microenvironment caused by pathogen challenge and dietary manipulations affect the *Gpr164* gene expression⁴. These findings, including ours, indicate a positive correlation between GPR164 and gut microbial diversity (new lines 402-416).”

Comments: Figure 6: FITC-dextran study in organoids is nice. It complements the Caco TER data in Figure 1 and demonstrates that the butyrate effect on barrier integrity is GPR164-dependent.

It would be helpful to have a quantitative measure, or at least present data from more than one organoid per condition, to increase the robustness of this finding.

Responses: According to the referee's suggestions, we quantified the fluorescence

intensities of FITC-dextran (new Fig. 6E).

Comments: Butyrate increases TJ protein expression and TER in Caco cells as well, but this was not done. If GPR164 KO Caco cells do not similarly respond to butyrate, this would significantly increase the rigor of the author's conclusion that GPR164 is necessary for butyrate-mediated epithelial barrier function is GPR164 dependent.

Responses: According to the referee's suggestion, we examined the effect of butyrate on *Claudin-3* mRNA expression and TER value when Caco-2 cells were treated with palmitate. *Claudin-3* mRNA expression was decreased when control KO cells were treated with palmitate, which tended to be restored by pretreatment of butyrate (new Fig. 6D). However, butyrate failed to restore *Claudin-3* expression in *OR51E1* KO cells treated with palmitate (new Fig. 6D). Notably, decreased TER value under stimulation with palmitate was partially restored by pretreatment with butyrate only in control KO cells (new Fig. 6E). These data strengthen our findings. We greatly appreciate the referee's helpful suggestion.

Dear Dr. Ikeda

Thank you for the submission of your revised manuscript to EMBO reports. Since my colleague Achim is currently out of office, I have temporarily taken over the handling of your manuscript. We have received the full set of referee reports that is copied below.

As you will see, the referees feel that the revision has significantly strengthened your manuscript, but they raise a number of remaining concerns, that I ask you to address in the manuscript and in a point by point response.

Browsing through the manuscript myself, I noticed a few editorial things that we need before we can proceed with the official acceptance of your study. Please also provide a point-by-point response to the editorial points to speed up further checks.

- Please place the 5 keywords below the Abstract.

- Please update the references to the alphabetical Harvard style. The abbreviation 'et al' should be used if there are more than 10 authors. You can download the respective EndNote file from our Guide to Authors https://endnote.com/style_download/embo-reports/

- You have provided a separate file with author contributions. We now use CRediT to specify the contributions of each author in the journal submission system, therefore this file is not needed. Please make sure that the author contributions in our online manuscript tracking system are correct and up-to-date. The information you specified in the system will be automatically retrieved and typeset into the article. You can enter additional information in the free text box provided, if you wish.

- Please correct the callout for Supplementary Table EV1 should be corrected to Table EV1.

- Please move the Appendix Supplementary Methods to the main methods section. All materials and methods must be part of the main manuscript file.

- Reagent and Tools table: Please remove the Instructions paragraph.

- In the Author Checklist you note that you have deposited human clinical and genomic datasets in a public repository. As far as I could see, the RNA seq data were obtained exclusively from mice, or did I miss anything? Could you please clarify? If human clinical data were deposited we would need information on the authority granting data collection and patient consent.

- The source data .xls files need to be split into one .xls file per figure panel. Each Figure folder needs subfolders for the panels containing only the source data for this panel.

- The microscopy images you listed in the Source Data checklist seem to be missing, as is the data for Fig. 3A, 3E, 4D, 5F and 6A. Could you please provide these as well?

- Please provide higher resolution images as the microscopy images appear pixelated.

- Our production/data editors have asked you to clarify several points in the figure legends (see below). Please incorporate these changes in the manuscript and return the revised file with tracked changes with your final manuscript submission.

A) Statistical test information. Only p-values that are actually shown in the figure panel(s) should (and must) be defined in the legends, all others should be removed from (or added to) the legend. Moreover, we ask for the specification of exact p-values, unless the p-values are small ($p < 0.0001$), which can be reported as inequalities:

1. Please define the annotated p values ****/****/**/* as well as provide the exact p-values for the same in the legend of figure 2F as appropriate.

2. Please note that the exact p values are not provided in the legends of figures 1B, C, D, E, F; 2B, C, D, E, G, H; 3B, E, F, G; 4A, B; 5A, B, C, D, E, G, H; 6B-D; EV1 F, EV2 A-C; EV3 A, C; EV5

3. Please indicate the statistical test used for data analysis in the legends of figures 2F, 3D

B) Replicates and error bars:

4. Please note that the box plots need to be defined in terms of minima, maxima, centre, bounds of box and whiskers, and percentile in the legends of figures 4A, B

5. Please note that the error bars are not defined in the legends of figures 1G, 2F

- Please add MW markers to the Western blot in Figure 1G.

- The legend for Table EV1 should be removed from the manuscript and included above the table in the Excel file.

- As a standard procedure we edit the title and abstract to make them more amenable to our non-specialist readership. Please

see my suggestions copied below my signature.

- Finally, EMBO Reports papers are accompanied online by

A) a short (1-2 sentences) summary of the findings and their significance,

B) 2-3 bullet points highlighting key results and

C) a schematic summary figure that provides a sketch of the major findings (not a data image).

Please provide the summary figure as a separate file in PNG or JPG format at a size of 550x300-600 pixels (width x height).

Please note that the size is rather small and that text needs to be readable at the final size. Please send us this information along with the revised manuscript.

With kind regards,

=====

Referee #1:

The authors have addressed most of my questions and there are just some minor things left before this manuscript can be accepted for publication.

Line 111: it is only the Paneth cells in the small intestine that migrate in an opposite direction. The sentence makes it sound like they all do. Please rephrase.

Line 119-121: "Although butyrate is rapidly used as fuel in the colon, intake of non-digestive carbohydrates or the progression of colorectal cancer leads to an increased accumulation of butyrate, which allows butyrate to act as a signaling molecule for FFARs and an HDAC inhibitor." I think this sentence is a bit unclear. How would cancer cause an accumulation of butyrate? I don't see how the reference supports this statement. Please explain how cancer increases butyrate or remove that part.

Line 221-222: Please revise the sentence to "The mucus layer, which mainly is composed of O-glycosylated Muc2 mucin, serves as a physical barrier against luminal pathogens in the colon." The original sentence reads like the mucus is only free mucin O-glycans.

Figure 3B: The two figures do not have the same magnification. The WT image is a higher magnification. Please replace with images with the same magnification. Please add information to the methods section regarding how the mucus layer was quantified including how many images that were analysed.

EVFig1G: The immunostaining of GRP164 is not very convincing. I know that I asked for this but I suggest that you exclude these images. I am well aware that antibody stainings not always work and I do not think this image looks very convincing even though the GPR164^{-/-} tissue is empty.

EVFig4B: Similar to the previous comments, I do not think that the Lgr5 staining pattern is very convincing as Lgr5 is a stem cell marker and the staining does not specifically stain the bottom of the crypts. I do not think that the manuscript benefits from these images.

EVFig4E: I find it strange that you do not have any CD4⁺ cells in the WT tissue since there is plenty of CD4⁺ cells also in the healthy colon. Do you have an explanation to why you do not see any CD4⁺ immune cells?

I would like the authors to add information to the methods section regarding how the fluorescence signal was quantified in the organoid experiments and how many organoids/experiments that were analysed.

If these comments are addressed, the manuscript is suitable for publication.

Referee #2:

I have read the revised Ms and the authors' responses to the reviewers' comments. The manuscript is now presented more clearly and the reviewer's comments have been used to improve specific aspects of the presentation. The authors' responses are detailed clear and helpful. The manuscript provides excellent data for anyone interested in the intestinal barrier.

Referee #3:

My overall sense of the original and revised manuscript remains the same - the claim that Gpr164 operates in the intestine to affect WNT-dependent phenotypes is reasonably (now better) supported. Suggested mechanisms of action however remain inconclusively tested - though the paper has been improved. Along with more reasonable revised interpretation of the data.

But here, I suggest that the paper has uncovered a likely mechanism of action explaining all phenotypes observed in mice lacking GPR164 - by the pathogenic hyperactivation of Wnt signaling.

I suggest this should be expanded upon - especially with respect to the disconnect between the increased goblet cell numbers and defective mucus layer observed in KO mice. Such a disconnect implies a defect in goblet cell biology and function (i.e. goblet cell differentiation). As I understand the field, Wnt signaling is diminished and inhibited as cells migrate up from crypt base to luminal surface. This diminishes the proliferative stem cell phenotype and enables cellular differentiation - perhaps (likely) affecting development of the mature specialized goblet cell phenotype that accounts for the mucus layer. Such a defect in cell differentiation/maturation would also explain the dysbiosis/inflammation observed as gut microbes live off the mucus layer. I also suggest the same defect in cellular differentiation might explain the defects in intercellular junctions and barrier function observed.

Regardless, revising manuscript some way so as to address more clearly the glaring disconnect in goblet cell expression and mucus layer development is needed. The one sentence added is not satisfying.

=====

Suggested abstract and title:

The free fatty acid receptor GPR164 maintains intestinal homeostasis and barrier function

GPR164 is a novel free fatty acid receptor, activated by both short-chain fatty acids and medium-chain fatty acids, and expressed throughout the gastrointestinal tract. Although GPR164 is reported to be involved in the release of gut hormones, the physiological functions of this receptor in the maintenance of intestinal homeostasis remain unclear. In this study, we explore the role of GPR164 in regulating intestinal barrier function using mice lacking Gpr164 gene (Gpr164^{-/-}). A loss-of-function mutation in GPR164 promotes cell proliferation and disrupts the intestinal barrier function in both Caco-2 cells and mice. Genome-wide RNA-seq analysis reveals that GPR164 deletion causes aberrant Wnt/ β -catenin signaling, and the intraperitoneal injection of the Wnt/ β -catenin inhibitor PNU-74654 ameliorates intestinal hyperproliferation, differentiation and barrier permeability phenotypes of Gpr164^{-/-} mice. Gpr164^{-/-} mice also exhibit gut microbial dysbiosis and inflammation. Thus, our findings uncover the pivotal role of GPR164 in the maintenance of intestinal homeostasis through regulating the barrier function.

The Senior Editor
EMBO Reports

Dear Dr. Rembold

Thank you very much for your e-mail concerning our manuscript (EMBOR-2025-61597V1) entitled "The pivotal role of a novel free fatty acid receptor GPR164 in the intestinal barrier function". We revised our manuscript to meet the reviewer's comments, and herewith enclosed the revised manuscript. We addressed all the points raised by all reviewers, as described below (Point-by point response). Also, we modified the revised manuscript in accordance with the editor's requirements.

We hope that this manuscript is now acceptable for publication in *EMBO Reports*. We are most grateful to you and reviewers for helpful comments.

Sincerely yours,

Takako Ikeda
Ikuo Kimura

Laboratory of Molecular Neurobiology, Graduate School of Biostudies, Kyoto University.

Sakyo-ku, Kyoto 606-8501, Japan.

Tel; (+81) 75-753-4547

email: Takako Ikeda (ikedata.kakako.2r@kyoto-u.ac.jp)

Ikuo Kimura (kimura.ikuo.7x@kyoto-u.ac.jp)

Response to the editor's comments
(Point-by point response)

Thank you for the submission of your revised manuscript to EMBO reports. Since my colleague Achim is currently out of office, I have temporarily taken over the handling of your manuscript. We have received the full set of referee reports that is copied below.

As you will see, the referees feel that the revision has significantly strengthened your manuscript, but they raise a number of remaining concerns, that I ask you to address in the manuscript and in a point by point response.

Browsing through the manuscript myself, , I noticed a few editorial things that we need before we can proceed with the official acceptance of your study. Please also provide a point-by-point response to the editorial points to speed up further checks.

Comments: Please place the 5 keywords below the Abstract.

Responses: According to the editor's comment, we placed the keywords below the Abstract.

Comments: Please update the references to the alphabetical Harvard style. The abbreviation 'et al' should be used if there are more than 10 authors. You can download the respective EndNote file from our Guide to Authors

https://endnote.com/style_download/embo-reports/

Responses: According to the editor's comment, we updated the references.

Comments: You have provided a separate file with author contributions. We now use CRediT to specify the contributions of each author in the journal submission system, therefore this file is not needed. Please make sure that the author contributions in our online manuscript tracking system are correct and up-to-date. The information you specified in the system will be automatically retrieved and typeset into the article. You can enter additional information in the free text box provided, if you wish.

Responses: I appreciate your helpful and kind comments.

Comments: Please correct the callout for Supplementary Table EV1 should be corrected to Table EV1.

Responses: According to the editor's comment, we corrected this (new line 503).

Comments: Please move the Appendix Supplementary Methods to the main methods section. All materials and methods must be part of the main manuscript file

Responses: According to the editor's comment, the Appendix Supplementary Methods were moved to the main methods section.

Comments: Reagent and Tools table: Please remove the Instructions paragraph.

Responses: According to the editor's comment, the Instructions paragraph in the Reagent and Tools table was removed.

Comments: In the Author Checklist you note that you have deposited human clinical and genomic datasets in a public repository. As far as I could see, the RNA seq data were obtained exclusively from mice, or did I miss anything? Could you please clarify? If human clinical data were deposited we would need information on the authority granting data collection and patient consent.

Responses: Our RNA seq data were obtained exclusively from mice, so that is our fault. We have corrected it in the Data availability section of the Author Checklist.

Comments: The source data .xls files need to be split into one .xls file per figure panel. Each Figure folder needs subfolders for the panels containing only the source data for this panel.

Responses: According to the editor's comment, the source data .xls files were split into one .xls file per figure panel.

Comments: The microscopy images you listed in the Source Data checklist seem to be missing, as is the data for Fig. 3A, 3E, 4D, 5F and 6A. Could you please provide these as well?

Responses: According to the editor's comment, we uploaded the data for Fig. 3A, 3E, 4D, 5F and 6A as the Source Data.

Comments: Please provide higher resolution images as the microscopy images appear pixelated.

Responses: According to the editor's comment, we uploaded the image data with higher resolution as the Source Data.

Comments: Our production/data editors have asked you to clarify several points in the figure legends (see below). Please incorporate these changes in the manuscript and return the revised file with tracked changes with your final manuscript submission.

A) Statistical test information. Only p-values that are actually shown in the figure panel(s) should (and must) be defined in the legends, all others should be removed from (or added to) the legend. Moreover, we ask for the specification of exact p-values, unless the p-values are small ($p < 0.0001$), which can be reported as inequalities:

1. Please define the annotated p values *****/***/**/* as well as provide the exact p-values for the same in the legend of figure 2F as appropriate.

Responses: According to the editor's comment, we defined the annotated p values and provided the exact p values in the legend of figure 2F.

Comments: 2. Please note that the exact p values are not provided in the legends of figures 1B, C, D, E, F; 2B, C, D, E, G, H; 3B, E, F, G; 4A, B; 5A, B, C, D, E, G, H; 6B-D; EV1 F, EV2 A-C; EV3 A, C; EV5

Responses: According to the editor's comment, we provided the exact p values in the legends of all you mentioned.

Comments: 3. Please indicate the statistical test used for data analysis in the legends of figures 2F, 3D

Responses: According to the editor's comment, we showed the statistical test used for

data analysis in the indicated figure legends.

Comments: B) Replicates and error bars:

4. Please note that the box plots need to be defined in terms of minima, maxima, centre, bounds of box and whiskers, and percentile in the legends of figures 4A, B

Responses: According to the editor's comment, we added the information for the box plot (Maximum, Minimum, Median, Percentile) in the legends.

Comments: 5. Please note that the error bars are not defined in the legends of figures 1G, 2F

Responses: According to the editor's comment, we defined the error bars in the legends of figures 1G, 2F.

Comments: Please add MW markers to the Western blot in Figure 1G.

Responses: According to the editor's comment, we added MW markers to the Western blot in Figure 1G.

Comments: The legend for Table EV1 should be removed from the manuscript and included above the table in the Excel file.

Responses: According to the editor's comment, we removed the legend for Table EV1 from the manuscript and added it above the table in the Excel file.

Comments: As a standard procedure we edit the title and abstract to make them more amenable to our non-specialist readership. Please see my suggestions copied below my signature.

Responses: In accordance with the editor's suggestions, we changed the title and rephrased the abstract.

Comments: Finally, EMBO Reports papers are accompanied online by

A) a short (1-2 sentences) summary of the findings and their significance,

B) 2-3 bullet points highlighting key results and

C) a schematic summary figure that provides a sketch of the major findings (not a data

image).

Please provide the summary figure as a separate file in PNG or JPG format at a size of 550x300-600 pixels (width x height). Please note that the size is rather small and that text needs to be readable at the final size. Please send us this information along with the revised manuscript.

Responses: In accordance with the editor's demands, we prepared and uploaded them.

Response to the reviewers' comments
(Point-by point response)

Referee #1:

Comments: The authors have addressed most of my questions and there are just some minor things left before this manuscript can be accepted for publication.

Line 111: it is only the Paneth cells in the small intestine the migrates in an opposite direction. The sentence makes it sound like they all do. Please rephrase.

Responses: According to the referee's comment, we rephrased it (deleted the word "bidirectionally") as described below.

"These differentiated cells migrate along the crypt axis, and are removed by apoptosis upon reaching the top of crypt (new lines, 111-112)."

Comments: Line 119-121: "Although butyrate is rapidly used as fuel in the colon, intake of non-digestive carbohydrates or the progression of colorectal cancer leads to an increased accumulation of butyrate, which allows butyrate to act as a signaling molecule for FFARs and an HDAC inhibitor." I think this sentence is a bit unclear. How would cancer cause an accumulation of butyrate? I don't see how the reference support this statement. Please explain how cancer increases butyrate or remove that part.

Responses: According to the referee's comment, we added the reference and the sentences to explain how butyrate is accumulated in colon cancer as described below.

"Although butyrate is rapidly used as fuel in the colon, intake of non-digestive carbohydrates leads to an increased accumulation of butyrate because butyrate is produced by the microbial fermentation of non-digestive carbohydrates as substrates. In addition, the progression of colorectal cancer also increases the level of butyrate¹². The colon cancer cells preferentially utilize glucose rather than butyrate as fuel, and the remaining butyrate is accumulated in colon. In both cases, increased butyrate concentration allows butyrate to act as a signaling molecule for FFARs and an HDAC inhibitor (new lines, 118-125)."

Comments: Line 221-222: Please revise the sentence to "The mucus layer, which mainly is composed of O-glycosylated Muc2 mucin, serves as a physical barrier against luminal pathogens in the colon." The original sentence reads like the mucus is only free mucin O-glycans.

Responses: According to the referee's comment, we rephrased the sentence as described below.

"The mucus layer, which is composed of complex glycans including Muc2 mucin O-glycan, serves as a physical barrier against luminal pathogens in the colon¹⁵(new lines, 219-220)."

Comments: Figure 3B: The two figures do not have the same magnification. The WT image is a higher magnification. Please replace with images with the same magnification. Please add information to the methods section regarding how the mucus layer was quantified including how many images that were analysed.

Responses: According to the referee's comment, we replaced the images with the same magnification. We apologize for our mistakes. Also, we added the information regarding how the mucus layer was quantified and how many images were analyzed as described below.

"The cross-sections were obtained from six independent individuals, and relative thickness of mucin layer was measured using ImageJ software (new lines, 568-569)."

Comments: EVFig1G: The immunostaining of GRP164 is not very convincing. I know that I asked for this but I suggest that you exclude these images. I am well aware that antibody stainings not always work and I do not think this image looks very convincing even though the GPR164^{-/-} tissue is empty.

Responses: In accordance with the referee's comment, we deleted these images.

Comments: EVFig4B: Similar to the previous comments, I do not think that the Lgr5 staining pattern is very convincing as Lgr5 is a stem cell marker and the staining does not specifically stain the bottom of the crypts. I do not think that the manuscript benefits

from these images.

Responses: According to the referee's comment, we deleted these images and rephrased the sentences as described below.

“On the contrary, mRNA expression of *Lgr5* was reduced in *Gpr164*^{-/-} mice, implying the possibility that colonic epithelial stem cells in the crypt cannot be received an appropriate Wnt signaling or the proportion of *Lgr5* positive stem cells was reduced because of the increased number of total epithelial cells (new lines, 287-290).”

Comments: EVFig4E: I find it strange that you do not have any CD4⁺ cells in the WT tissue since there is plenty of CD4⁺ cells also in the healthy colon. Do you have an explanation to why you do not see any CD4⁺ immune cells?

Responses: As the referee mentioned, we observed CD4⁺ cells in the WT tissue as well as in the KO tissue. We replaced the data because the fluorescence intensity of CD4 shown in old Fig. EV4E was weak. We appreciate your insightful comment.

Comments: I would like the authors to add information to the methods section regarding how the fluorescence signal was quantified in the organoid experiments and how many organoids/experiments that were analysed.

If these comments are addressed, the manuscript is suitable for publication.

Responses: According to the referee's comment, we added the information regarding how the fluorescence signal was quantified in the organoid experiments and how many organoids were analyzed as described below.

“The fluorescence intensity within the organoids isolated from three independent individuals was measured using ImageJ software (new lines, 642-644)”

Referee #2:

Comments: I have read the revised Ms and the authors' responses to the reviewers' comments. The manuscript is now presented more clearly and the reviewer's comments have been used to improve specific aspects of the presentation. The authors' responses are detailed clear and helpful. The manuscript provides excellent data for anyone interested in the intestinal barrier.

Responses: We greatly appreciate the referee's kind comment.

Referee #3:

Comments: My overall sense of the original and revised manuscript remains the same - the claim that *Gpr164* operates in the intestine to affect WNT-dependent phenotypes is reasonably (now better) supported. Suggested mechanisms of action however remain inconclusively tested - though the paper has been improved. Along with more reasonable revised interpretation of the data.

But here, I suggest that the paper has uncovered a likely mechanism of action explaining all phenotypes observed in mice lacking GPR164 - by the pathogenic hyperactivation of Wnt signaling.

I suggest this should be expanded upon - especially with respect to the disconnect between the increased goblet cell numbers and defective mucus layer observed in KO mice. Such a disconnect implies a defect in goblet cell biology and function (i.e. goblet cell differentiation). As I understand the field, Wnt signaling is diminished and inhibited as cells migrate up from crypt base to luminal surface. This diminishes the proliferative stem cell phenotype and enables cellular differentiation - perhaps (likely) affecting development of the mature specialized goblet cell phenotype that accounts for the mucus layer. Such a defect in cell differentiation/maturation would also explain the dysbiosis/inflammation observed as gut microbes live off the mucus layer. I also suggest the same defect in cellular differentiation might explain the defects in intercellular junctions and barrier function observed.

Regardless, revising manuscript some way so as to address more clearly the glaring disconnect in goblet cell expression and mucus layer development is needed. The one sentence added is not satisfying.

Responses: According to the referee's comment, we added the sentences to address the relationship between cell differentiation/maturation and mucus layer development as described below. We greatly appreciate your constructive comments.

“The mucus layer is important for a physical barrier against luminal pathogens in the colon. In this study, we observed strong staining for alcian blue-PAS in colon of *Gpr164*^{-/-} mice (Fig. 3A), which is consistent with the results obtained by RNA-seq

analysis (Fig. 4B). These results suggest that loss of GPR164 results in increased differentiation into goblet cells. However, thickness of mucus layer in *Gpr164*^{-/-} mice was reduced as compared to WT mice (Fig. 3B). Considering that GPR164 plays an important role in regulating differentiation of colonic epithelial cells, this receptor may affect the function of goblet cells. Indeed, increased stored mucins were observed in goblet cells of *Gpr164*^{-/-} mice (Fig. 3A), suggesting the possibility that the function of goblet cells such as mucus secretion is impaired in *Gpr164*^{-/-} mice. The colon has a two-layered mucus, inner and outer mucus layer. The commensal bacteria colonize the outer mucus layer which provides the ideal habitat as a nutritional source and microbial adhesion sites. Thus, a defect in the outer mucus layer alters the microbial composition, leading to gut dysbiosis. The inner mucus layer serves as a physical barrier to prevent pathogenic infection. The inner mucus defects allow bacteria to reach colonic epithelial cells, which causes gut inflammation by stimulating host immune response. Notably, our study revealed that loss of *Gpr164* resulted in gut microbial dysbiosis and inflammation (Figs. 3C-F). Therefore, our findings imply that GPR164 maintains mucus barrier function by regulating goblet cell maturation (new lines 395-413).”

Dr. Takako Ikeda
Kyoto University
Graduate School of Biostudies
Sakyo-ku
Konoe-cho
Kyoto 606-8502
Japan

Dear Dr. Ikeda,

Thank you for the submission of your further revised manuscript to our editorial offices. Going through the manuscript and your p-b-p-response, I consider the remaining points by the referees and the editorial requests as adequately addressed.

I am thus very pleased to accept your manuscript for publication in the next available issue of EMBO reports. Thank you for your contribution to our journal.

Yours sincerely,
